# SpARQ: Outlier-Free SpeechLM with Fast Adaptation and Robust Quantization

## Abstract

We propose **SpARQ** (outlier-free **Sp**eechLM for Fast **A**daptation and **R**obust **Q**uantization) to tackle the outlier problem of **Speech** and **L**anguage multi-modal **M**odels (**SpeechLM**s). Our primary observation is that outliers stemming from cross-modal (speech and text) low-rank adaptation and post-training quantization stages affect the performance of the current SpeechLMs. Methodologically, **SpARQ** leverages a pretrained language model as its foundation, substituting the traditional attention layer with a novel stabilized outlier-free layer. This modification eliminates outliers typically arising during cross-modal low-rank adaptation and post-training quantization. The model is then fine-tuned on multi-modal data using proposed outlier-free architecture, allowing it to handle textLM, speechLM, ASR, and TTS tasks through a unified interface while maintaining compatibility with parameters adapted from standard pretrained LLMs. Consequently, on the OPT-1.3b model, the proposed framework achieves relative performance improvements: 41% in cross-modal low-rank adaptation and 45% in post-training quantization, along with a 1.33x training speedup. We benchmark it against state-of-the-art low-rank adaptation and post-training quantization methods.

## 1 Introduction

SpeechLM leverages pretrained language models to significantly enhance speech recognition and synthesis technologies, transforming our ability to understand and generate natural speech (Nguyen et al., 2024). Traditional Automatic Speech Recognition (ASR) (Park et al., 2019) and Text-to-Speech (TTS) (Hayashi and Watanabe, 2020) systems rely on specialized components like feature extraction, acoustic or linguistic modeling, and waveform synthesis. In contrast, as illustrated in Figure 1, SpeechLM integrates these modalities within a single Large Language Model (LLM) framework, using a unified token space for both speech and text as input and output. This integration enables ASR, TTS, speech generation (SpeechLM), and text generation (textLM) tasks within the same framework (Maiti et al., 2023; Yang et al., 2023), leveraging the robust capabilities of text-based language models to enhance performance, efficiency, and adaptability in speech-related applications.

However, efficiently adapting LLMs to handle speech remains a major challenge (Mehrish et al., 2023). While some speech and audio understanding models (Chu et al., 2023; Wang et al., 2023a; Gong et al., 2023; 2024) have employed efficient techniques like low-rank adaptation or freezing the entire LLM when integrating new inputs, these methods have not been applied to the SpeechLM framework for generating outputs in both speech and text modalities (Maiti et al., 2023; Yang et al., 2023; Zhang et al., 2024; Défossez et al., 2024). The fundamental differences between text and speech modalities—such as variable sequence lengths and data representations—complicate this adaptation process. Consequently, there is a pressing need for methods to achieve the efficiency gains of low-rank adaptation and post-training quantization for the SpeechLM framework.

Studies have shown that transformer-based models often focus on less informative tokens, resulting in inefficiencies due to outliers (Clark et al., 2019; Kovaleva et al., 2019; Zhao et al., 2024; Huang et al., 2024). In multimodal frameworks, outliers present in pretrained LLMs and multimodal generation tasks (Zhang et al., 2020) exacerbate these challenges, and as demonstrated in (Crabbé et al., 2024), multimodal models exhibit a substantial number of outliers in their attention mechanisms. During low-rank adaptation, outliers from the pretrained model mix with those introduced by the multimodal data, leading to distorted outputs and diminished performance. Similarly, in post-training quantization, outliers inherited from the pretrained model negatively impact the quality of the final output.

Figure 1: **Illustration of the SpeechLM System.** The language model is trained using a next-token prediction objective. Speech tokens are generated by encoding speech with the HuBERT encoder and can be decoded back to their original modality. We train the SpeechLM models on a combination of sequences: text-only, speech-only, speech-to-text (ASR), and TTS. In the figure, we demonstrate an ASR example that uses special tokens "ST" (start of text) and "GS" (generate speech) alongside text and speech tokens.

To address these issues and enable efficient fine-tuning and post-training quantization (e.g., SmoothQuant (Xiao et al., 2023)) in transformer-based models with multimodal inputs, we propose a novel stabilized outlier-free layer as a substitute for the conventional attention mechanism (Vaswani et al., 2017). Unlike prior work that applies outlier mitigation during pretraining, we apply our approach during the domain adaptation fine-tuning procedure for low-rank adaptation (LoRA) and post-training quantization on pretrained LLMs. Our design builds upon the outlier-free Hopfield layer introduced by (Hu et al., 2024a), which has been proven to effectively identify and filter out outliers.

Our approach enhances transformer-based pretrained Language Learning Models (LLMs) through the integration of a specialized stabilization module within the outlier-free layer. This innovative design enables direct parameter adaptation of existing pretrained vanilla LLMs, eliminating the need for training an outlier-efficient model from scratch. Such an approach preserves the inherent capabilities of pretrained transformers while substantially reducing computational overhead. The framework's effectiveness is further amplified through the incorporation of low-rank adaptation techniques, such as LoRA (Hu et al., 2021), alongside post-training quantization methods. These implementations yield marked performance improvements compared to conventional transformer-based models. Through this enhancement strategy, we have developed a solution that not only optimizes efficiency but also increases practical applicability, making it valuable in resource-constrained environments where training capabilities may be limited.

**Contributions.** We propose Stablized Outlier-free **Sp**eechLM with Fast **A**daptation and **R**obust **Q**uantization (**SpARQ**) for multi-modal speech-text tasks. Our contributions are as follows:

- **Methodological Innovation:** We present a stabilized outlier-free layer as a replacement for the standard transformer attention mechanism in the SpeechLM model. This layer mitigates outliers that emerge during text-based LLM adaptation to multi-modal generation, improving performance in speech applications while maintaining stability through LoRA and quantization processes. The outlier-free layer works in concert with LoRA to minimize trainable parameters in LLM-to-SpeechLM adaptation, resulting in enhanced quantized model performance.

- **Efficient Adaptation of Pretrained LLMs:** The stabilization module in our outlier-free layer enables direct parameter initialization from transformer-based LLMs to outlier-free SpeechLM. This method matches the performance of full LLM retraining under outlier-free conditions. The integration with LoRA delivers superior results in ASR and TTS tasks compared to standard transformer-based SpeechLM architectures. Our approach optimizes resource utilization while maintaining performance, making it ideal for deployment in computationally constrained environments.

- **Empirical Validation:** Our work presents the first successful implementation of this approach in a multi-modal SpeechLM framework that combines speech and text generation through LoRA, QLoRA, and post-training quantization techniques. Using the Open Pretrained Transformer (OPT) as our foundation, we conducted comprehensive evaluations of our method's performance and efficiency. Benchmarks against current low-rank adaptation methods and post-training quantization techniques demonstrate the framework's effectiveness. The results show relative performance gains of 41% in cross-modal post-training quantization and 45% in low-rank adaptation compared to standard frameworks.

**Organization.** Section 2 includes a description of recent work in SpeechLM and Low-rank adaptation fine-tuning. Section 3 details the system settings of SpeechLM and the outlier-efficient architecture in our framework. Section 4 presents the numerical evaluation results demonstrating the efficiency of our framework. Section 5 provides a conclusion and discusses future research directions.

## 2 RELATED WORK

**Discrete Speech Representation.** Recent advances in Self-Supervised Learning (SSL) for speech have improved our ability to derive meaningful representations from raw audio data. These advancements enable the extraction of discrete speech tokens from SSL models like HuBERT (Hsu et al., 2021) and w2v-BERT (Chung et al., 2021) for various speech-processing applications. These SSL models generate semantic tokens by clustering learned features, capturing the linguistic content of the speech. This approach transforms speech into pseudo-text, facilitating applications in speech-based natural language understanding and generation. Models like HuBERT cluster continuous speech features into discrete tokens that represent phonetic or sub-word units, improving the accuracy and efficiency of high-level speech processing tasks, such as TTS (Hayashi and Watanabe, 2020) , speech-to-speech translation (S2ST) (Lee et al., 2021), and ASR (Park et al., 2019).

**Speech and Text LMs.** Joint modeling of speech and text has gained significant attention in recent studies. Initial approaches (Ao et al., 2021; Chen et al., 2022) proposed learning shared speech-text representations with separate encoders and decoders, requiring alignment losses for cross-modal transfer. Recent methods employ a single model for multiple tasks. For example, SpeechGPT (Zhang et al., 2023) combines audio generation with textLMs, PolyVoice applies speechLM to S2ST (qian Dong et al., 2023), SpiritLM (Nguyen et al., 2024) excels in speech and expressive speech generation, also adapted for related speech tasks, and Voxtlm (Maiti et al., 2024) conducts speech/text generation along with ASR and TTS. We utilize textually pre-trained OPT (Zhang et al., 2022) for better initialization inspired by (Maiti et al., 2024; Hassid et al., 2024) and leverage different speech tokens, ensuring full reproducibility of our work.

**Low-Rank Adaptation and Post Training Quantization.** Low-Rank Adaptation (Xin et al., 2024) and Post Training Quantization (PTQ) (Gholami et al., 2022) are essential techniques for reducing the memory footprint and latency of large foundation models (Bommasani et al., 2021), i.e. huge transformer-based models. Those large foundation models play a crucial role not only in machine learning area but also in a huge scientific area, such as (Zhou et al., 2024) for genomics, (Wang et al., 2023b; Wu et al., 2023) for financial, and (Maiti et al., 2024) for speech. However, large foundation models are resource-intensive. Low-Rank Adaptation and PTQ play crucial roles in deploying these large models on edge devices with limited resources. Significant contributions have been made in the area of Low-Rank Adaptation (Dettmers et al., 2024; Li et al., 2023; Hu et al., 2021) and PTQ (Zafrir et al., 2019; Dettmers et al., 2022). However, these methods typically do not address the outlier problem during the Low-Rank Adaptation and quantization processes, as highlighted by (Hu et al., 2024a). To tackle this issue, we propose the Outlier-Free Layer to manage outliers effectively during both the Low-Rank Adaptation and quantization processes.

## 3 METHODOLOGY

This section introduces the Stabilized Outlier-Free SpeechLM system, which adapts LLMs to speech and text modalities by replacing standard transformer layers with a stabilized Outlier-Free Layer. This layer mitigates challenges posed by outliers during cross-modal low-rank adaptation and post-training quantization, addressing instabilities encountered with existing outlier layers.

### 3.1 SPEECHLM SETUP

Our goal is to model both speech and text modalities within a unified framework. To achieve this, we convert continuous speech signals into discrete tokens $s_i \in \mathcal{V}_{\text{sp}}$ using a speech tokenizer, such as a HuBERT model with $k$-means clustering. These speech tokens are integrated with text tokens $t_i \in \mathcal{V}_{\text{txt}}$ to form a shared vocabulary $\mathcal{V}_{\text{joint}} = \mathcal{V}_{\text{txt}} \cup \mathcal{V}_{\text{sp}}$. We combine tokenized training data from multiple tasks to create a joint dataset and train a subword model (e.g., BPE or SentencePiece) on this data. We model the probability of a text utterance $T = (t_i)$ as $p(T) = \prod_i p(t_i \mid t_1, \cdots, t_{i-1})$. Similarly, continuous speech signals are converted into discrete tokens $S = (s_i)$ and modeled likewise. The joint probability of speech and text tokens $J = (j_i \in \mathcal{V}_{\text{joint}})$ is expressed as $p(J) = \prod_i p(j_i \mid j_1, \cdots, j_{i-1})$. To enable the model to handle multiple tasks within this unified framework, we incorporate four special tokens to indicate tasks: start of speech (SS)/text (ST) and generate speech (GS)/text (GT), following (Maiti et al., 2023). In SpeechLM and textLM tasks, GS and GT appear at the beginning of sentences. For

**a). Vanilla SpeechLM Framework**     **b). Proposed SpARQ Framework**     **c). Proposed Layer**

Figure 2: **Illustration of the SpARQ Framework and Outlier-free Layer.** The vanilla SpeechLM framework employs a pretrained LLM as its backbone for processing cross-modal inputs (text and speech). However, this approach often results in output outliers, as illustrated in Figure a). To address this issue, we propose SpARQ, which replaces the original softmax activation function with an outlier-free layer. Figure b) demonstrates SpARQ's effectiveness in reducing outliers. The structure of our outlier-free layer, detailed in Figure c), comprises a max-shift normalization followed by a softmax_1 activation function.

ASR and TTS tasks, sentences start with SS or ST and use GT or GS as prediction start tokens. This setup enables unified next-token prediction across all tasks.

**Modality Fusion.** SpeechLM accepts both speech and text inputs within the shared vocabulary $\mathcal{V}_{\text{joint}}$, formed by combining text characters and discrete speech tokens obtained using $k$-means clustering on pre-trained HuBERT features (Hsu et al., 2021), as described earlier. For speech synthesis, we use HiFi-GAN (Kong et al., 2020), incorporating x-vector (Snyder et al., 2018) as the speaker embedding, both pretrained on LJSpeech (Ito., 2017). Additionally, to enhance contextual information and reduce sequence length, we apply sentencepiece subword modeling (Kudo and Richardson, 2018) within $\mathcal{V}_{\text{joint}}$ (Radford et al.; Song et al., 2020; Chang et al., 2023). During training, we employ teacher forcing in an autoregressive manner. At each timestep $i$, SpARQ predicts the distribution $\widehat{p}^i = \text{SpARQ}(j_1, \ldots, j_{i-1})$., and the cross-entropy loss is computed as:

$$L_{CE}(p_i, \widehat{p}^i) = - \sum_{c=1}^{|\mathcal{V}_{\text{joint}}|} p_i(c) \log \widehat{p}^i(c),$$

where $p_i$ is the true probability distribution. During inference, the model predicts the next tokens conditioned on the input as $p(\cdot \mid \text{condition})$. For TTS, given a text utterance $T_{\text{test}}$, it predicts speech tokens $\widehat{S}$. For ASR, given speech tokens $S_{\text{test}}$, it predicts text $\widehat{T}$. For speech or text continuation, it predicts continued speech tokens $\widehat{S}$ or text $\widehat{Y}$ based on the respective prefixes. The model is trained on these cross-modality tasks using the task-specific tokens introduced earlier.

### 3.2 OUTLIER FREE ARCHITECTURE

The Outlier-free layer is our proposed solution to mitigate challenges posed by outliers during the model's cross-modal low-rank adaptation and post-training quantization processes. The proposed outlier efficient architecture is shown in Figure 2.

**Problem Setup.** We consider a multimodality speech-text framework where the input is a $d$-dimensional speech-text sequence of length $L$, represented as a matrix $X \in \mathbb{R}^{d \times L}$ for compatibility with the transformer architecture. The matrix $X$ is then processed by transformer layers, which are composed of multiple self-attention mechanisms followed by feed-forward neural networks.

**Motivational Example.** Studies by (Clark et al., 2019) and (Kovaleva et al., 2019) find that in BERT, certain tokens such as delimiters and punctuation marks receive disproportionately high attention weights compared to other tokens. Additionally, as stated in (Bondarenko et al., 2024) and (Hu et al., 2024a), tokens, despite having low informational value, tend to attract significant attention probabilities. Such behaviors lead to excessive computational and memory demands during the model training process and contribute to significant performance degradation during model quantization. As demonstrated in (Crabbé et al., 2024), multimodal models exhibit a substantial number of outlier in their attention. We use this observation as a starting point by considering the following attention mechanism:

$$\text{Output} = \text{Residual}(\text{Softmax}(XW_q XW_k^\mathsf{T}/\sqrt{d})XW_v + X). \tag{3.1}$$

With the given formula, if the attention input $X$ already holds sufficient information, the transformer should prevent from making updates. However, due to the characteristics of the $\text{Softmax}$ function, tokens with smaller values receive disproportionately high attention weights. This results in a broad distribution of attention scores and introduces outliers that negatively impact the model's performance. Furthermore, cross-modal inputs (text and speech) may inherently possess distinct statistical characteristics. This can complicate the normalization of attention scores and potentially

introduce additional outliers that affect the model's performance. We provide a visualization of the outliers in the SpeechLM system in Appendix A.

Given this challenge, we would like to develop a transformer architecture that can efficiently solve the challenge in Equation (3.1). To achieve this, we propose an outlier-free layer consisting of a stabilized layer and a memory-associated activation function. There is a huge amount of work proposed to reduce the outlier challenges in each of the model stages, pre-training (Hu et al., 2024a), fine-tuning (Chen et al., 2024; Hu et al., 2024b), and inference (Xiao et al., 2023; Bondarenko et al., 2024). following (Miller, 2023; Hu et al., 2024a), we use outlier-free memory associated ($\mathrm{Softmax}_1$) function (as shown in 3.2) for input vector $S$ to tackle the challenge of the outliers in LLMs. We term this architecture outlier-free Transformer, and briefly comment on it. Firstly, our proposed architecture is applied to the attention mechanism in the transformer layer with the activation function $\mathrm{Softmax}_1$. Secondly, we introduce a stabilized layer to ensure numerical stability in the $\mathrm{Softmax}_1$ function, forming a stabilized outlier-free layer as:

$$\mathrm{Softmax}_1(\mathrm{S}) = \frac{\exp(\mathrm{S})}{1 + \sum_{i=1}^{L} \exp(\mathrm{S_i})}, \text{ where } \mathrm{S} = \mathrm{S} - \max(\mathrm{S}). \tag{3.2}$$

With the model size increasing, the $\mathrm{Softmax}_1$ function in (Hu et al., 2024a) brings gradient problem due to the numerical instability in $\mathrm{Softmax}_1$ function (Alman and Song, 2024). Recent works (Hu et al., 2024b; Jiang et al., 2023) provide theory support for the essential of the stabilized layer, and we have tried different stabilized methods as shown in Table 7.

## 3.3 Integrating the Outlier-Free Architecture into SpeechLM

We accommodate the outlier-free architecture into the SpeechLM framework to create our proposed SpARQ system. This integration enables efficient adaptation of pre-trained LLMs to the multi-modal speech-text setting and facilitates effective post-training quantization.

Two main challenges arise in this process. First, adapting text-only pre-trained LLMs to a multi-modal environment involves handling an expanded vocabulary $\mathcal{V}_{\mathrm{joint}}$, longer sequences due to speech tokens—which are about five times more numerous than text tokens—and the presence of outliers that hinder both low-rank adaptation and post-training quantization. This complexity makes applying parameter-efficient training methods like LoRA and quantization techniques challenging. Second, since the pre-trained LLMs are based on vanilla transformer architectures, replacing the standard transformer layers with the outlier-free layers can introduce instability. The model parameters are not initially optimized for the new architecture, necessitating a stabilization mechanism to ensure training proceeds smoothly without significant perturbations.

**Cross-Modal Adaptation.** To address the first challenge, we utilize our stabilized outlier-free layers to mitigate the impact of outliers during adaptation. By effectively handling outliers, our approach enables both efficient low-rank adaptation techniques like LoRA and robust post-training quantization methods such as SmoothQuant. This allows the model to manage the expanded vocabulary and longer sequences more effectively, facilitating parameter-efficient training and quantization. Consequently, we reduce trainable parameters and computational overhead compared to full fine-tuning methods used in previous works (Maiti et al., 2023; Nguyen et al., 2024), and improve quantization performance.

**Stabilized Outlier-Free Adaptation.** For the second challenge, we incorporate a stabilization module within the outlier-free layers. This addition mitigates instability caused by layer replacement, allowing us to initialize directly from pre-trained vanilla LLM parameters without extensive pretraining specific to the outlier-free architecture. As a result, we achieve effective cross-modal learning with improved training stability and significant performance gains in ASR and TTS tasks compared to traditional transformer-based SpeechLM systems.

## 3.4 Theoretical Justifications

Building on the theoretical advantages of the outlier-free transformer reported by (Hu et al., 2024a), we provide two additional justifications for applying LoRA to the outlier-free transformer.

**Expressiveness.** We provide the expressive guarantee of Low-Rank Adaption for transformer model with $\mathrm{Softmax}_1$. We identify the conditions for the existence of low-rank adapters for exact adaptation. We summarize our main findings in the following informal theorem.

**Proposition 3.1** (Expressiveness of LoRA $\mathrm{Softmax}_1$ (Informal Version of Theorem C.1))**.** Let $f^{\star}$ and $\bar{f}$ denote the pretrained and target transformer models, respectively, using the Stabilized Outlier-Free

layer (3.2) as the backbone. Under mild non-singularity and LoRA-rank conditions, there exist low-rank adapters such that the modified model $f$ is exactly equal to $\bar{f}$.

**Training Efficiency.** We find that the attention weights are concentrated on significant tokens, enabling less training time cost during fine-tuning compared to the vanilla version, as shown in Table 3. We provide a theoretical justification for why we observe improved LoRA training efficiency.

**Proposition 3.2** (Fast LoRA Requires Proper Normalization (Informal Version of Proposition C.1)). Let $X \in \mathbb{R}^{d \times L}$ be the input sequence, and let $r$ denote the rank of the LoRA adapters for a pretrained transformer model. Sub-quadratic time-efficient LoRA training up to a precision of $\epsilon = O((\log L)^{-4})$ is achievable if the following conditions hold: (i) long sequence setting with $d = O(\log L)$, (ii) mild rank $r < d$, and (iii) proper normalization of the input and model weights.

**Remark 3.1.** Our outlier-free layer ensures proper normalization of the model weights. Our stabilization technique ensures proper normalization of the model input.

We defer more comprehensive theoretical justifications to Appendix C.

## 4 Experimental Studies

In this section, we conducted experiments to validate our proposed framework's effectiveness, comparing its performance against state-of-the-art methods described in (Maiti et al., 2023). Our evaluation utilized three sizes of OPT pretrained models: OPT-125m, OPT-350m, and OPT-1.3b.

**Computational Resource.** We perform all experiments using 4 NVIDIA A100 GPU with 80GB of memory and a 24-core Intel(R) Xeon(R) Gold 6338 CPU operating at 2.00GHz. Our code is developed in PyTorch and utilizes the Hugging Face Transformer Library for experimental execution.

**Models.** Following (Maiti et al., 2023), we validate our method using different sizes of Open Pretrained Transformer (OPT) models. We employ the same BPE model as (Maiti et al., 2023) for four tasks: textLM, speechLM, ASR, and TTS. The training procedure for all OPT models follows (Hu et al., 2024a).

**Datasets.** Our experiments utilize four datasets across different tasks. For textLM, we use Librispeech (Panayotov et al., 2015), comprising 40 million text utterances. SpeechLM employs LibriLight (LL) (Kahn et al., 2020), which contains 60,000 hours of audiobook recordings from 7,000 speakers, totaling 12 million utterances. For ASR tasks, we use the English Multilingual Librispeech (MLS) dataset (Pratap et al., 2020). TTS experiments are conducted using LibriTTS (LT) (Zen et al., 2019) and VCTK (VC) (Veaux et al., 2017) datasets.

**Evaluation Metrics.** We employ specific metrics for each task in our evaluation process. For speech and text generation tasks, we assess models with identical vocabulary sizes using perplexity (PPL). In ASR tasks, we utilize Word Error Rate (WER) as our primary metric. For TTS evaluation, we employ Hifi-gan (Kong et al., 2020) as the vocoder and measure intelligibility using the WER derived from whisper decoding results. Notably, lower scores in these metrics indicate superior performance. Additionally, to evaluate the model performance on the next-token prediction, we report the model's next-token accuracy in the ablation study Section 4.3.

### 4.1 Post-Training Quantization (PTQ)

To evaluate the efficiency of our method, we employ the proposed Outlier-free layer in all OPT models (Zhang et al., 2022) as a substitute for the standard attention layer as described in (Vaswani et al., 2017). We utilize the pre-trained OPT model checkpoints with the SpARQ framework (Hu et al., 2024a), and fine-tune the model at full rank according to the approach outlined in (Maiti et al., 2023). We then evaluate them on the test datasets using FP16 (16-bit floating-point) and perform state-of-the-art Post-training Quantization (PTQ) methods to assess the performance drop from FP16. We conduct each evaluation 3 times with different random seeds and present the average and standard deviation for each metric. Since the standard deviations of all results are less than 2%, we omit them in our result tables.

**Baselines.** Following (Maiti et al., 2023), we validate our method with three different quantization methods: SmoothQuant (Xiao et al., 2023), AffineQuant (Ma et al., 2024), and OmniQuant (Shao et al., 2023). We consistently apply the same hyperparameters detailed in their respective studies. Specifically, the hyperparameters for SmoothQuant follow the guidelines set forth in (Xiao et al., 2023). Similarly, for AffineQuant, we adhere to the parameters described in (Ma et al., 2024). Lastly,

Table 1: **Comparing SpARQ with Vanilla Framework in a Post-Training Quantization (PTQ) setting.** Experiments were conducted across three quantization methods (SmoothQuant, AffineQuant, OmmiQuant) on a low bit weight and activation quantization setting – weight 4 bits and activation 4 bits (W4A4). Evaluation metrics included Text PPL, SpeechLM PPL, ASR WER, and TTS WER. We assessed the average performance drop across these four tasks post-quantization. Results show SpARQ consistently outperforms Vanilla framework, exhibiting smaller performance drops when applying low bit quantization methods, demonstrating its superior efficiency in PTQ settings.

| Model | Method | #Bits | Quantization Method | TextLM PPL ($\downarrow$) | SpeechLM PPL ($\downarrow$) | ASR WER ($\downarrow$) | TTS WER ($\downarrow$) | Avg Performance Drop ($\downarrow$) |
|---|---|---|---|---|---|---|---|---|
| OPT-125m | Vanilla | W16/A16 | - | 22.56 | 59.42 | 12.40 | 12.08 | - |
| | | W4/A4 | SmoothQuant | 45.23 | 96.87 | 52.31 | 48.79 | 197.31% |
| | | W4/A4 | AffineQuant | 31.25 | 80.19 | 29.44 | 28.34 | 86.37% |
| | | W4/A4 | OmmiQuant | 31.28 | 80.21 | 31.98 | 29.55 | 94.04% |
| | SpARQ | W16/A16 | - | 22.70 | 59.45 | 12.61 | 12.11 | - |
| | | W4/A4 | SmoothQuant | 37.14 | 84.55 | 35.32 | 36.73 | **112.05%** |
| | | W4/A4 | AffineQuant | 26.11 | 68.42 | 14.33 | 15.71 | **18.52%** |
| | | W4/A4 | OmmiQuant | 26.12 | 68.63 | 14.53 | 16.01 | **19.63%** |
| OPT-350m | Vanilla | W16/A16 | - | 13.13 | 43.10 | 8.42 | 17.56 | - |
| | | W4/A4 | SmoothQuant | 36.74 | 75.38 | 40.17 | 70.53 | 233.37% |
| | | W4/A4 | AffineQuant | 27.28 | 66.31 | 36.84 | 40.83 | 157.92% |
| | | W4/A4 | OmmiQuant | 27.85 | 67.83 | 37.54 | 41.37 | 162.73% |
| | SpARQ | 16/16A | - | 13.47 | 43.34 | 9.81 | 17.31 | - |
| | | W4/A4 | SmoothQuant | 23.48 | 62.17 | 36.22 | 40.83 | **130.71%** |
| | | W4/A4 | AffineQuant | 22.82 | 51.74 | 25.78 | 28.44 | **78.97%** |
| | | W4/A4 | OmmiQuant | 22.83 | 52.08 | 26.11 | 29.15 | **81.05%** |
| OPT-1.3b | Vanilla | W16/A16 | - | 12.62 | 41.33 | 8.00 | 18.73 | - |
| | | W4/A4 | SmoothQuant | 36.74 | 87.46 | 48.96 | 53.15 | 249.63% |
| | | W4/A4 | AffineQuant | 24.31 | 61.74 | 43.68 | 32.47 | 165.33% |
| | | W4/A4 | OmmiQuant | 24.43 | 62.38 | 44.52 | 33.03 | 169.34% |
| | SpARQ | W16/16A | - | 12.95 | 42.48 | 8.25 | 12.07 | - |
| | | W4/A4 | SmoothQuant | 23.83 | 58.33 | 32.27 | 33.12 | **146.72%** |
| | | W4/A4 | AffineQuant | 20.81 | 48.84 | 22.78 | 25.46 | **90.68%** |
| | | W4/A4 | OmmiQuant | 20.88 | 48.97 | 23.58 | 26.83 | **96.15%** |

for OmniQuant, the hyperparameters are in line with those specified in (Shao et al., 2023). This approach ensures that our evaluations are based on standardized settings, allowing for accurate comparisons and assessments of each quantization method.

**Results.** Table 1 demonstrates SpARQ's superior performance over standard training frameworks in W4A4 post-training quantization scenarios using state-of-the-art PTQ methods. Under AffineQuant, the vanilla framework experiences performance drops of 86.37%, 107.78%, and 165.33% for OPT-125m, OPT-350m, and OPT-1.3b respectively. SpARQ reduces these declines to 18.52%, 64.96%, and 90.68%. For OPT-1.3b, SpARQ achieves a 45% relative improvement in average W4A4 quantization performance, highlighting its robustness in low-bit quantization for large models. Additional Weight-8bit-Activation-8bit (W8A8) quantization results appear in Appendix D.

## 4.2 LOW-RANK ADAPTATION METHODS

To show the effectiveness of the SpARQ framework in the cross-modal low-rank adaptation process, we compare SpARQ with the vanilla training framework across two different LoRA techniques.

**LoRA Methods.** We compare SpARQ with the vanilla training framework across two different LoRA methods: LoRA (Hu et al., 2021) and QLoRA (Dettmers et al., 2024). We fine-tune the model without using any adaptation method as the full-rank baseline, following the setting in (Maiti et al., 2023). For the LoRA method, following (Hu et al., 2021), we fine-tune the model with low-rank adaptations using a rank of 128 and an alpha value of 256. For the QLoRA method, following (Dettmers et al., 2024), we fine-tune the model with quantized low-rank adaptations, maintaining the same rank and alpha value as specified in LoRA, but using Int8 (Dettmers et al., 2022) quantization instead of 4-bit NormalFloat (NF4) (Dettmers et al., 2024).

**Results.** In Table 2, our results demonstrate the effectiveness of SpARQ in cross-modal low-rank adaptation. As a result, the SpARQ framework offers a performance improvement of relative 41% in low-rank adaptation over the vanilla framework in OPT-1.3b. Specifically, SpARQ exhibits a smaller performance drop across all three model sizes. However, we observe a significant decline in performance when using LoRA, with WER dropping from 8.00% to 46.92% for ASR and from

Table 2: **Comparing SpARQ with Vanilla Framework in a Low-Rank Adaptation Setting.** We conduct experiments on SpARQ with vanilla attention across two Low-Rank Adaptation methods (LoRA, QLoRA). The evaluation metrics include Text Perplexity (PPL), SpeechLM PPL, and Word Error Rate (WER) in Automatic Speech Recognition (ASR) and Text-to-Speech (TTS). We also measure the average performance drop after low-rank adaptation to assess the efficiency of SpARQ in the low-rank adaptation setting. In most configurations, SpARQ results in better fine-tuning performance compared to vanilla attention.

| Model | Method | Low-Rank Adaptation Method | TextLM PPL ($\downarrow$) | SpeechLM PPL ($\downarrow$) | ASR WER ($\downarrow$) | TTS WER ($\downarrow$) | Average Performance Drop ($\downarrow$) |
|---|---|---|---|---|---|---|---|
| OPT-125m | Vanilla | Full | 22.56 | 59.42 | 12.40 | 12.08 | - |
| | | LoRA | 25.69 | 62.16 | 12.39 | 15.47 | 11.61% |
| | | QLoRA | 25.97 | 62.43 | 12.86 | 15.02 | 12.06% |
| | SpARQ | Full | 22.58 | 59.46 | 12.61 | 12.11 | - |
| | | LoRA | 25.77 | 62.23 | 12.56 | 11.80 | **3.96%** |
| | | QLoRA | 25.77 | 62.23 | 13.42 | 12.46 | **7.03%** |
| OPT-350m | Vanilla | Full | 13.13 | 43.10 | 8.42 | 17.56 | - |
| | | LoRA | 17.87 | 51.65 | 93.91 | 97.06 | 233.98% |
| | | QLoRA | 18.07 | 51.50 | 76.02 | 98.63 | 211.49% |
| | SpARQ | Full | 13.47 | 43.34 | 9.81 | 17.31 | - |
| | | LoRA | 17.71 | 51.13 | 18.52 | 75.18 | **106.68%** |
| | | QLoRA | 16.64 | 48.34 | 22.30 | 83.36 | **123.93%** |
| OPT-1.3b | Vanilla | Full | 12.62 | 41.33 | 8.00 | 18.73 | - |
| | | LoRA | 17.14 | 50.22 | 46.92 | 94.53 | 237.12% |
| | | QLoRA | 17.87 | 51.43 | 87.64 | 43.11 | 369.20% |
| | SpARQ | Full | 12.95 | 42.48 | 8.20 | 12.07 | - |
| | | LoRA | 16.83 | 49.51 | 8.25 | 74.21 | **140.44%** |
| | | QLoRA | 17.54 | 50.99 | 43.18 | 94.23 | **290.16%** |

18.73% to 94.53% for TTS in vanilla OPT-1.3b. One possible explanation for this substantial performance drop is that larger models have greater difficulty forgetting the original knowledge learned from text and adapting to new knowledge in a different modality, especially with a limited number of fine-tuning epochs. This anomaly aligns with the findings of (Von Oswald et al., 2019; Ramasesh et al., 2021). On the other hand, although the SpARQ models experience a performance drop with low-rank adaptation, the decline is minimal for ASR (8.20% to 8.25%) and remains better than the vanilla version for TTS (12.07% to 74.21%) for OPT-1.3b.

### 4.3 ABLATION STUDY

This section presents experimental results across three key areas. We analyze outlier differences between vanilla and SpARQ frameworks, evaluate our training methodology's contribution to efficiency, and examine stabilization methods' impact on model convergence. These analyses reveal critical factors driving our framework's performance improvements.

**Outlier in the SpeechLM System.** Our quantitative assessment of outlier effects compares vanilla and SpARQ frameworks using maximum infinity norm and average kurtosis metrics (lower values indicate better performance). The evaluation spans three tasks and three low-rank adaptation methods. Results in Figure 3 show SpARQ's consistent reduction of outlier effects across all scenarios. This improvement manifests in both uni-modal (text or speech)

Table 3: **Comparison of the Full Fine-tuning Training Time (Per Epoch) between the Vanilla Framework and SpARQ.**

| Model | Method | Training Time per Epoch |
|---|---|---|
| OPT-350m | Vanilla | 74 mins |
| | SpARQ | **58 mins** |
| OPT-1.3b | Vanilla | 84 mins |
| | SpARQ | **63 mins** |

and cross-modal (text and speech) tasks, maintaining consistency across all low-rank adaptation methods. The findings demonstrate SpARQ's enhanced stability in managing diverse input modalities and adaptation techniques, yielding superior SpeechLM performance.

**Efficient Training.** To evaluate the efficiency of our proposed framework, SpARQ, we compared its training speed with the vanilla training framework. We measured the training time per epoch for both methods in two different model sizes. As shown in Table 3, SpARQ consistently accelerates training across various model sizes. Specifically, SpARQ achieves a 1.28x speedup for OPT-350m and a 1.33x speedup for OPT-1.3b compared to the vanilla framework.

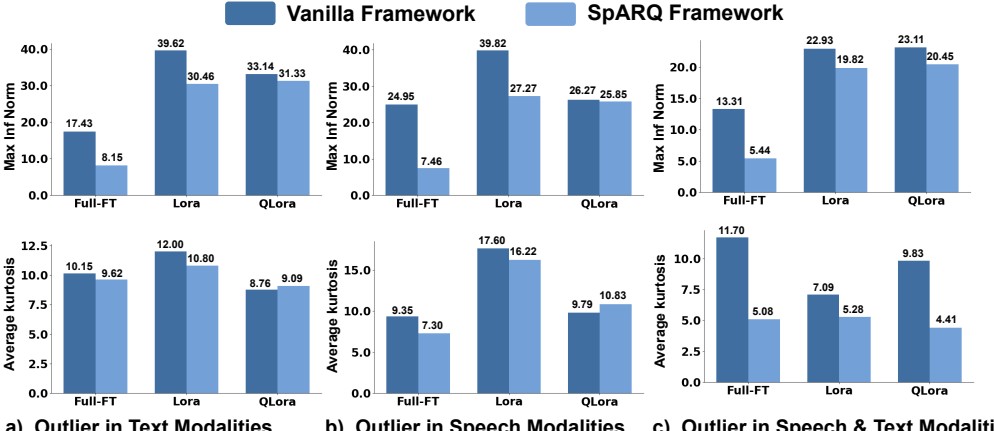

Figure 3: **Outlier Comparison between Vanilla and SpARQ Framework.** We evaluate the average kurtosis and maximum infinity norms of outliers for (a) text modalities using TextLM data, (b) speech modalities using SpeechLM data, and (c) speech-text cross-modalities using ASR data. Outliers are measured at the sentence level for all scenarios. For ASR, we include both speech and corresponding ground-truth text as input.

**Different Stabilized Methods.** To investigate the efficacy of various input vector stabilization techniques, we conducted a systematic comparative analysis of three normalization strategies: $L_1$ normalization, max-shift normalization, and mean-centering normalization. Our experimental results, presented in Table 7, reveal that max-shift normalization uniquely achieves model stabilization and resolves

Table 4: **Comparison of Different Stabilized Methods.**

| Model | Stabilized Method | Val Accuracy (%) |
|---|---|---|
| | N/A | N/A |
| OPT-1.3b | L_1 | N/A |
| | Max-Shift | **34.6** |
| | Mean-Centering | N/A |

gradient-related issues. In contrast, both $L_1$ and mean-centering normalizations proved ineffective, resulting in numerical instabilities during the training process. These instabilities are denoted as 'N/A' in our results, indicating that the model encountered NaN (Not a Number) losses, rendering training infeasible. These findings underscore the critical role of appropriate normalization in maintaining model stability and highlight the superior performance of max-shift normalization in our framework.

## 5 DISCUSSION AND CONCLUSION

We introduce SpARQ, an outlier-robust multi-modal foundation model for speech-text tasks. SpARQ addresses the computational challenges arising from outlier effects in modality fusion and cross-modality adaptation of SpeechLM. Our approach mitigates outlier impacts in transformer-based models and enhances both low-rank adaptation and post-training quantization performance. SpARQ demonstrates significant improvements over existing methods, achieving a 41% relative performance gain in cross-modal low-rank adaptation (Section 4.2) and 45% in quantization (Section 4.1)

**Limitations and Future Work.** SpARQ currently has two key limitations. First, it does not support LoftQ and other SVD-based low-rank adaptation methods that operate on weight matrices. Second, due to computational constraints, SpARQ uses neither the 6.7B parameter OPT model nor other large decoder-based models for pretraining. " Future work will expand SpARQ to incorporate SVD-based methods and evaluate performance with larger decoder architectures, including 3B LLama2 and 6.7B OPT models. Additionally, the framework's focus on reducing computational demands could inadvertently amplify biases inherited from pre-trained models. Further investigation is necessary to understand and address the impact of biases within our framework.

**Broader Impact.** Our methodology advances foundation model fine-tuning and inference through insights from associative memory models, enhancing both low-rank adaptation and post-training quantization. This solution enables edge computing deployment of large foundation models and resource-efficient fine-tuning. However, the approach may amplify existing training data biases, potentially disadvantaging underrepresented groups.

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

# Appendix

## A  VISUALIZATION OF THE OUTLIER IN THE SPEECHLM SYSTEM

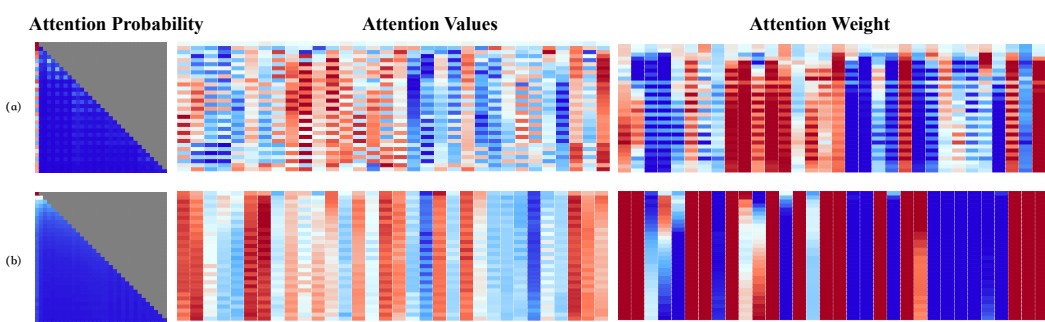

Figure 4: **Visualization of Attention Probability, Value and Weight in LoRA Funetuning.** We present a visualization of the attention probability, value, and weight for a cross-modality speech sample processed by the OPT-125m model. The visualization includes two scenarios: (a) the vanilla OPT-125m model, and (b) the Outlier-Free OPT-125m model (Hu et al., 2024a). Additionally, we visualize the model's hidden representations in the last hidden layers during the LoRA fine-tuning process and scale up all heatmaps from range 0 (blue) to 1 (red). In the vanilla model, we observe that the attention probability is distributed across various tokens rather than being concentrated on specific, significant tokens. This dispersion causes the model to expend effort on unnecessary tokens during fine-tuning, leading to performance degradation and resources inefficiency. In contrast, the Outlier-Free model shows a more focused attention distribution, which helps in reducing the computational effort required for fine-tuning by concentrating on the significant tokens.

As shown in Figure 4, we use visualization to highlight the challenges posed by outliers in transformer-based models during the fine-tuning period. We visualize the model's hidden representations in the last hidden layers during the LoRA fine-tuning process. In the figure, deeper shades of red indicate higher values of attention probability, value, and weight. Conversely, deeper shades of blue represent lower values. This color coding helps illustrate the concentration of attention and computational focus within the model. In the vanilla model, we observe that the attention probability is distributed across various tokens rather than being concentrated on specific, significant tokens. This dispersion can cause the model to expend effort on unnecessary tokens during fine-tuning, leading to performance degradation and resources inefficiency. In contrast, the Outlier-Free model shows a more focused attention distribution, which helps reduce the computational effort required for fine-tuning by concentrating on the significant tokens. Additionally, we find that the attention weight in Outlier-Free is higher for tokens with high attention values. This indicates that the model does not spend extra computational resources on less significant tokens, allowing the fine-tuning process to converge more efficiently.

## B  INFLUENCE OF ADAPTOR RANK

We conducted a detailed analysis to evaluate the performance of our proposed method across various ranks when applying Low-rank Adaptation (LoRA) in comparison to the standard, or vanilla, method. The results, as shown in Table 5, consistently demonstrate that our method outperforms the vanilla approach across all tested ranks. Notably, we observed that a rank of 256 delivers the optimal performance; therefore, we have chosen to use a rank of 256 for all subsequent low-rank adaptation experiments. Expanding on the observed results, it's important to understand

Table 5: Comparison of Different Ranks Using LoRA by Validation Accuracy

| Method | Fine-Tuning Method | Rank | Val Acc (%) |
|---|---|---|---|
| Vanilla | Full Fine-Tuning | N/A | **30.5** |
| SpARQ | Full Fine-Tuning | N/A | 30.2 |
| Vanilla | LoRA | 512 | 27.6 |
| SpARQ | LoRA | 512 | **27.8** |
| Vanilla | LoRA | 256 | 28.1 |
| SpARQ | LoRA | 256 | **28.9** |
| Vanilla | LoRA | 128 | 27.5 |
| SpARQ | LoRA | 128 | **27.5** |

why increasing the rank beyond 256 does not lead to further performance improvements. Typically, a higher rank in LoRA introduces more trainable parameters into the model. While this could potentially enhance the model's learning capacity, it also requires significantly more training epochs

for the model to effectively converge. This extended training process can introduce inefficiencies and practical limitations, which may negate the benefits of a higher parameter count. Thus, the rank of 256 strikes a balance between performance enhancement and computational efficiency, making it the most effective choice for our low-rank adaptation experiments.

## C  THEORETICAL ANALYSIS

### C.1  FAST LORA TRAINING OF $\text{Softmax}_1$ REQUIRES INPUT AND WEIGHT NORMALIZATION

Our theoretical justification for fast LoRA training on our outlier-free stabilized models an application of efficiency results of (Hu et al., 2024b). We follow the notation of (Hu et al., 2024b) in this section.

To present our results, we introduce the Strong Exponential Time Hypothesis (SETH) as a stronger form of the $\text{P} \neq \text{NP}$ conjecture.

**Hypothesis 1** (SETH). For every $\epsilon > 0$, there is a positive integer $k \geq 3$ such that $k$-SAT on formulas with $n$ variables cannot be solved in $\mathcal{O}(2^{(1-\epsilon)n})$ time, even by a randomized algorithm.

Formally, we formulate the *partial* adaptation of an attention head as the following LoRA loss.

**Definition C.1** (Adapting $W_Q$, $W_V$ of Generic Attention with LoRA). Let $\mathcal{D} = \{\left(X_i^{(K)}, X_i^{(Q)}, X_i^{(V)}\right), Y_i\}_{i=1}^N$ be a dataset of size $N$ with the triplet $X_i^{(K)}, X_i^{(Q)}, X_i^{(V)} \in \mathbb{R}^{L \times d}$ being the input and $Y_i \in \mathbb{R}^{L \times d}$ being the label. The problem of fine-tuning $W_Q$, $W_V$ a generic attention with LoRA with $\ell_2$ loss from dataset $\mathcal{D}$ is formulated as

$$\min_{\substack{B_Q, B_V \in \mathbb{R}^{d \times r} \\ A_Q, A_V \in \mathbb{R}^{r \times d}}} \mathcal{L}\left(W_K^\star, W_Q = W_Q^\star + \frac{\alpha}{r} B_Q A_Q, W_V = W_V^\star + \frac{\alpha}{r} B_V A_V\right) \tag{C.1}$$

$$:= \min_{\substack{B_Q, B_V \in \mathbb{R}^{d \times r} \\ A_Q, A_V \in \mathbb{R}^{r \times d}}} \frac{1}{2N} \sum_{i=1}^N \left\| \underbrace{\text{Softmax}_1\left\{ X_i^{(Q)} W_Q (W_K^\star)^\mathsf{T} \left(X_i^{(K)}\right)^\mathsf{T} \beta \right\}}_{(I)} \underbrace{X_i^{(V)} W_V}_{(II)} - Y_i \right\|_F^2.$$

To further simplify, we introduce $C_i^{(1)}, C_i^{(2)}, C_i^{(3)} \in \mathbb{R}^{L \times d}$ via

$$C_i^{(1)} := X_i^{(Q)} \frac{\alpha}{r} \in \mathbb{R}^{L \times d}, \quad C_i^{(2)} := X_i^{(K)} W_K^\star \in \mathbb{R}^{L \times d} \quad C_i^{(3)} := X_i^{(V)} W_V^\star. \tag{C.2}$$

Notably, $C_i^{(1)}, C_i^{(2)}, C_i^{(3)}$ are constants with respect to adapting (C.1) with gradient updates. To prove the hardness of Definition C.1 for both full gradient descent and stochastic mini-batch gradient descent, it suffices to consider adapting on a single data point. Thus, we deduce Definition C.1 to

$$\min_{\substack{B_Q \in \mathbb{R}^{d \times r} \\ A_Q \in \mathbb{R}^{r \times d}}} \mathcal{L}(B_Q, A_Q) = \min_{\substack{B_Q \in \mathbb{R}^{d \times r} \\ A_Q \in \mathbb{R}^{r \times d}}} \frac{1}{2} \left\| D^{-1} \exp\left\{ C^{(1)} \left(\overline{W}_Q^\star + B_Q A_Q\right) \left(C^{(2)}\right)^\mathsf{T} \right\} C^{(3)} - Y \right\|_F^2, \tag{C.3}$$

where $\overline{W}_Q^\star := r W_Q^\star / \alpha, D = \text{diag}\left(\exp\left\{ C^{(1)} \left(\overline{W}_Q^\star + B_Q A_Q\right) \left(C^{(2)}\right)^\mathsf{T} \right\} \mathbb{1}_L + \mathbf{I}_{L \times L}\right) \in \mathbb{R}^{L \times L}$.

We introduce the next problem to characterize all possible (efficient or not) gradient computation of optimizing (C.3). Let $Y[i, \cdot]$ and $Y[\cdot, j]$ be the $i$-th row and $j$-th column of $Y$, respectively.

**Problem 1** (Approximate LoRA Gradient Computation $\mathsf{ALoRAGC}(L, d, r, \epsilon)$). Given $C_i^{(1)}, C_i^{(2)}, C_i^{(3)}, Y_i \in \mathbb{R}^{L \times d}$. Let $\epsilon > 0$. Assume all numerical values are in $\log(L)$-bits encoding. Let $\mathcal{L}$ follow (C.3). The problem of approximating gradient computation of optimizing (C.3) is to find two matrices $\widetilde{G}_Q^{(A)} \in \mathbb{R}^{d \times r}$ and $\widetilde{G}_Q^{(B)} \in \mathbb{R}^{r \times d}$ such that $\max\left(\|\widetilde{G}_Q^{(B)} - \frac{\partial \mathcal{L}}{\partial B_Q}\|_\infty, \|\widetilde{G}_Q^{(A)} - \frac{\partial \mathcal{L}}{\partial A_Q}\|_\infty\right) \leq \epsilon$.

Finally we arrive our main result, the inefficient threshold for approximating gradient computation of (C.3). In the other words, we provide a inefficient threshold for adapting transformer-based models with LoRA in $L^{2-o(1)}$ (sub-quadratic) time. For convenience, we consider the special case Problem 1.

**Proposition C.1** (Efficient Threshold (Formal Version of Proposition 3.2, Modified from Theorem 5.1 of (Hu et al., 2024b))). Let $\kappa : \mathbb{N} \to \mathbb{N}$ by any function with $\kappa(L) = \omega(1)$ and $\kappa(L) = o(\log L)$. Let $\Gamma = O(\sqrt{\log L} \cdot \kappa(L))$. Assuming Hypothesis 1, there is no algorithm running in time $O(L^{2-\delta})$ for any constant $\delta > 0$ for $\mathsf{ALoRAGC}(L, d = O(\log L), r < d, \epsilon)$, i.e., Problem 1, subject to (C.3), even in the case where the input and weight matrices satisfy $\|X^{(K)} W_K^\star\|_\infty \leq \Gamma$, $\|\alpha X_i^{(Q)} B_Q A_Q / r\|_\infty \leq \Gamma$, $Y = 0$ and $\epsilon = O((\log L)^{-4})$.

**Remark C.1** (Remark 5.1 of (Hu et al., 2024b)). Proposition C.1 suggests a efficiency threshold for norm bound $\Gamma$ (norm of some composition of input $X$ and weights $W$s.) Specifically, Proposition C.1 implies that, only below this threshold, efficient (sub-quadratic) LoRA training of $\mathrm{Softmax}_1$-based transformer is possible.

## C.2 EXPRESSIVENESS GUARANTEE

In this section, we provide the expressive guarantee of Low-Rank Adaption for transformer model with $\mathrm{Softmax}_1$. Moreover, we identify the conditions for the existence of low-rank adapters.

We start with the definition of the target model $\overline{f}$ and the adopted model $f$.

**Definition C.2** (Definition of target model $\overline{f}$ and adopted model $f$). For any input $X \in \mathbb{R}^{D \times N}$, where $D$ denotes the dimension of token embedding and $N$ denotes the number of tokens. We consider a $H$-heads transformers $\mathrm{TF}_\theta$, consist of $L$-Transformer blocks and an output layer with parameter $\theta = \left( (W_{Ol}^h, W_{Vl}^h, W_{Kl}^h, W_{Ql}^h)_{h=1}^H, W_{1l}, W_{2l}, b_{1l}, b_{2l})_{l=1}^L, W_o \right)$. Specifically, we formulate it as

$$\text{Hidden layer: } \mathrm{Attn}(Z_{l-1}) = \sum_{h=1}^H \overline{W}_{Ol}^h \overline{W}_{Vl}^h \cdot \overline{Z}_{l-1} \cdot \mathrm{Softmax}_1(\overline{Z}_{l-1}^\top \overline{W}_{Kl}^{h\top} \overline{W}_{Ql}^h \overline{Z}_{l-1}),$$

$$Z_l = W_{2l} \cdot \mathrm{ReLU}(W_{1l} \cdot \mathrm{Attn}(Z_{l-1}) + b_{1l} \mathbf{1}_N^\top) + b_{2l} \mathbf{1}_N^\top$$

$$\text{OutputLayer}: \mathrm{TF}_\theta(X) = \mathrm{Softmax}_1(\mathrm{W_o Z_L}),$$

where we define $Z_0 := X$. Here, $W_{1l}^h, W_{Vl}^h, W_{Kl}^h, W_{Ql}^h \in \mathbb{R}^{D \times D}$ are weight matrices in $l$-th attention layer. Further, $W_{1l}, W_{2l} \in \mathbb{R}^{D \times D}$ are weight matrices and $b_{1l}, b_{2l}$ are the bias vectors in the $l$-th feedforward layer. Then we define the target model $\overline{f}$ and the adopted model $f$ are

$$\overline{f} := \mathrm{TF}_{\theta_T}, \quad \theta_T = \left( (\overline{W}_{Ol}^h, \overline{W}_{Vl}^h, \overline{W}_{Kl}^h, \overline{W}_{Ql}^h)_{h=1}^H, \overline{W}_{1l}, \overline{W}_{2l}, \overline{b}_{1l}, \overline{b}_{2l})_{l=1}^L, \overline{W}_o \right)$$

$$f := \mathrm{TF}_{\theta_A}, \quad \theta_A = ((W_{Ol}^h + \Delta W_{Ol}^h, W_{Vl}^h + \Delta W_{Vl}^h, W_{Kl}^h + \Delta W_{kl}^h, W_{Ql}^h + \Delta W_{Ql}^h)_{h=1}^H,$$

$$W_{1l} + \Delta W_{1l}, W_{2l} + \Delta W_{2l}, \widehat{b}_{1l}, \widehat{b}_{2l})_{l=1}^L, W_o + \Delta W_o).$$

Moreover, we define the best low-rank approximation for matrix $W$.

**Definition C.3** (Best Low-rank Approximation for $W$). For any matrix $W \in \mathbb{R}^{D \times D}$, the singular value decomposition (SVD) of $W$ is expressed as $W = UDV^\top$. Above $U, V \in \mathbb{R}^{D \times D}$ are orthonormal matrices and $D \in \mathbb{R}^{D \times D}$ is a diagonal matrix. Let the singular value of $W$ are denoted as $\sigma_1(W) \geq \ldots \geq \sigma_D(W) \geq 0$. When $d > D$, let $\sigma_d(W) = 0$. For any rank $r > 0$, we define

$$\mathrm{LRA}_r(W) := \sum_{i=1}^r \sigma_i(W) u_i v_i^\top,$$

where $u_i, v_i^\top$ are the $i$-th column of $U, V$, respectively.

According to (Eckart and Young, 1936; Mirsky, 1960), $\mathrm{LRA}_r(W)$ are the best rank-$r$ approximation in the Frobenius norm or the 2-norm of $W$. To present our results, we now introduce non-singularity assumption based on Definition C.3.

**Assumption C.1** (Non-Singularity). For a fixed $R \in [D]$, the weight matrices of both the target model, the pretrained model and the following matrices are non-singular, for all $r \in [R]$. Specifically,

$$W_{Kl}^{h\top} W_{Ql}^h + \mathrm{LRA}_r(\overline{W}_{Kl}^{h\top} \overline{W}_{Ql}^h - W_{Kl}^{h\top} W_{Ql}^h), \text{ for all } h \in [H] \text{ and } l = 1,$$

$$W_{Kl}^{h\top} W_{Ql}^h + \mathrm{LRA}_r(W_{2,l-1}^{-1\top} \overline{W}_{2,l-1}^\top \overline{W}_{Kl}^{h\top} \overline{W}_{Ql}^h \overline{W}_{2,l-1} W_{2,l-1}^{-1} - W_{Kl}^{h\top} W_{Ql}^h), \text{ for all } h \in [H], l \in [L]\backslash\{1\},$$

$$W_{Ol}^h W_{Vl}^h + \mathrm{LRA}_r(W_{1l}^{-1} \overline{W}_{1l} \overline{W}_{Ol}^h \overline{W}_{Vl}^h - W_{Ol}^h W_{Vl}^h), \text{ for all } h \in [H] \text{ and } l = 1,$$

$$W_{Ol}^h W_{Vl}^h + \mathrm{LRA}_r(W_{1l}^{-1} \overline{W}_{1l} \overline{W}_{Ol}^h \overline{W}_{Vl}^h \overline{W}_{2,l-1} W_{2,l-1}^{-1} - W_{Ol}^h W_{Vl}^h), \text{ for all } h \in [H] \text{ and } l \in [L]\backslash\{1\},$$

$$W_o W_{2L} + \mathrm{LRA}_r(\overline{W}_o \overline{W}_{2L} - W_o W_{2L}),$$

are non-singular, where LRA denotes the rank-$r$ approximation follows Definition C.3.

Under a non-singularity assumption (Assumption C.1), we apply another helper lemma from (Zeng and Lee, 2023) to construct the weight matrices in Theorem C.1.

**Lemma C.1** (Exactly represent target model, Lemma 7 of (Zeng and Lee, 2023)). Define error matrix $E := \overline{W} - \prod_{l=1}^{L} W_l$, and denote its rank by $R_E = \mathrm{rank}(E)$. For a given LoRA-rank $R \in [D]$, assume that all the weight matrices of the frozen model $(W_l)_{l=1}^L$, and $\prod_{l=1}^{L} W_l + \mathrm{LRA}_r(E)$ are non-singular for all $r \leq R(L-1)$. Then, the approximation error

$$\min_{\Delta W_l: \mathrm{rank}(\Delta W_l) \leq R} \left\| \prod_{l=1}^{L} (W_l + \Delta W_l) - \overline{W} \right\|_2 = \sigma_{RL+1} \underbrace{\left( \overline{W} - \prod_{l=1}^{L} W_l \right)}_{\text{Error matrix } E}$$

and the optimal solution to the matrix approximation problem satisfies $\prod_{l=1}^{L} (W_l + \Delta W_l) = \prod_{l=1}^{L} W_l + \mathrm{LRA}_{RL \wedge R_E}(E)$. Therefore, when $R \geq \lceil \frac{R_E}{L} \rceil$, we have $\prod_{l=1}^{L} (W_l + \Delta W_l) = \overline{W}$, implying $f \equiv \bar{f}$.

With Assumption C.1 and Lemma C.1, we show that for any input $X \in \mathbb{R}^{D \times D}$, there exists a adapted model $f$ capable of approximating target model $\bar{f}$ exactly, i.e, $f(X) = \bar{f}(X)$.

**Theorem C.1** (Express capability of transformers (Formal Version of )). Suppose LoRA-rank $R \in [D]$. Let Assumption C.1 hold. Define the rank-based functionality gap $G_i$ to i-th Stabilized Outlier-Free block ($i \in [L]$) or output layer ($i = L + 1$) as

$$G_i = \begin{cases} \max_h(\mathrm{rank}(\overline{W}_{Ki}^{h\top} \overline{W}_{Qi}^h - W_{Ki}^{h\top} W_{Qi}^h)) \vee \max_h(\mathrm{rank}(\overline{W}_{1i} \overline{W}_{Oi}^h \overline{W}_{Vi}^h - W_{1i} W_{Oi}^h W_{Vi}^h)), & i = 1, \\ \max_h(\mathrm{rank}(\overline{W}_{2,i-1}^\top \overline{W}_{Ki}^{h\top} \overline{W}_{Qi}^h \overline{W}_{2,i-1} - W_{2,i-1}^\top W_{Ki}^{h\top} W_{Qi}^h W_{2,i-1}), & \\ \vee \max_h(\mathrm{rank}(\overline{W}_{1i} \overline{W}_{Oi}^h \overline{W}_{Vi}^h \overline{W}_{2,i-1} - W_{1i} W_{Oi}^h W_{Vi}^h W_{2,i-1})), & 2 \leq i \leq L, \\ \mathrm{rank}(\overline{W}_o \overline{W}_{2L} - W_o W_{2L}), & i = L + 1. \end{cases}$$

If $R \geq \max_{i \in [L+1]} \lceil \frac{G_i}{2} \rceil$, then there exists low-rank adapters with rank lower than $R$ $((\Delta W_{Kl}^h, \Delta W_{Ql}^h, \Delta W_{Vl}^h, \Delta W_{Ol}^h)_{h=1}^H)_{l=1}^L, \Delta W_{2L}, \Delta W_o$ with other low-rank adapters set to $O$, and updated bias vectors $(\hat{b}_{1l}, \hat{b}_{2l})_{l=1}^L$, such that for any input $X \in \mathbb{R}^{D \times N}$, the adapted model $f$ exactly approximates target model $\bar{f}$, i.e., $f(X) = \bar{f}(X)$.

*Proof.* We build our proof on (Zeng and Lee, 2023).

First, we ensure that, for each Stabilized Outlier-Free block, the output from the first feedforward layer in the target model matches that in the adapted model. Then, we select an appropriate output layer weight matrix to complete the proof.

We define $\overline{H}_l \in \mathbb{R}^{D \times N}$ and $\overline{Z}_l \in \mathbb{R}^{D \times N}$ as the intermediate and final outputs of the $l$-th transformer block in the target model $\overline{f}$, respectively. For any $l \in [L]$, they are formulated as

$$\overline{H}_l := \text{ReLU}(\overline{W}_{1l}(\sum_{h=1}^{H} \overline{W}_{Ol}^h \overline{W}_{Vl}^h \cdot \overline{Z}_{l-1} \cdot \text{Softmax}_1(\overline{Z}_{l-1}^\top \overline{W}_{Kl}^{h\top} \overline{W}_{Ql}^h \overline{Z}_{l-1})) + \overline{b}_{1l}\mathbf{1}_N^\top),$$

$$\overline{Z}_l := \overline{W}_{2l}\overline{H}_l + \overline{b}_{2l}\mathbf{1}_N^\top.$$

Correspondingly, we introduce $\widehat{H}_l$ and $\widehat{Z}_l$ to denote the intermediate output of the first feedforward layer and the final output of the $l$-th Stabilized Outlier-Free block for the adapted model $f$,

$$\widehat{H}_l = \text{ReLU}(W_{1l}(\sum_{h=1}^{H}(W_{Ol}^h + \Delta W_{Ol}^h)(W_{Vl}^h + \Delta W_{Vl}^h) \cdot \widehat{Z}_{l-1} \tag{C.4}$$

$$\cdot \text{Softmax}_1(\widehat{Z}_{l-1}^\top(W_{Kl}^h + \Delta W_{Kl}^h)^\top(W_{Ql}^h + \Delta W_{Ql}^h)\widehat{Z}_{l1}) + \widehat{b}_{1l}\mathbf{1}_N^\top),$$

$$\widehat{Z}_l = W_{2l}\widehat{H}_l + \widehat{b}_{2l}\mathbf{1}_N^\top, \tag{C.5}$$

for any $l \in [L]$. Note that $\overline{Z}_0 = \widehat{Z}_0 = X$. Next, we inductively construct the adapter weight matrices $((\Delta W_{Ol}^h, \Delta W_{Vl}^h, \Delta W_{Kl}^h, \Delta W_{Ql}^h)_{h=1}^H, \widehat{b}_{1l}, \widehat{b}_{2l})_{l=1}^L$ such that $\widehat{H}_l = \overline{H}_l$ for all $l \in [L]$. We then select the low-rank adapters for $W_{2L}$ and the $W_o$ to approximate the output of the target model. For unmentioned low-rank adapters, we set them as $O$.

**When $l = 1$.** To achieve $\widehat{H}_l = \overline{H}_l$ for all $X$, the following conditions must be satisfied:

- Bias Vector: $\widehat{b}_{1l} = \overline{b}_{1l}$,

- Query and Key: $(W_{Kl}^h + \Delta W_{Kl}^h)^\top(W_{Ql}^h + \Delta W_{Ql}^h) = \overline{W}_{Kl}^{h\top}\overline{W}_{Ql}^h$,

- Value and First Feedforward Layer: $(W_{Ol}^h + \Delta W_{Ol}^h)(W_{Vl}^h + \Delta W_{Vl}^h) = W_{1l}^{-1}\overline{W}_{1l}\overline{W}_{Ol}^h\overline{W}_{Vl}^h$,

It is simple to check that we only need to set $\widehat{b}_{1l} = \overline{b}_{1l}$ to, and select rank-$R$ or lower matrices $\Delta W_{Kl}^h, \Delta W_{Ql}^h, \Delta W_{Ol}^h, \Delta W_{Vl}^h$ as suggested by Lemma C.1. This ensures $\widehat{H}_l = \overline{H}_l$ for $l = 1$.

**When $l > 1$.** For the cases $l = 2, \ldots, L$, we assume the induction hypothesis holds for $l - 1$, which is $\widehat{H}_{l-1} = \overline{H}_{l-1}$. We let $\widehat{b}_{2,l-1} = W_{2,l-1}\overline{W}_{2,l-1}^{-1}\overline{b}_{2,l-1}$, then it holds,

$$\widehat{Z}_{l-1} = W_{2,l-1}\overline{W}_{2,l-1}^{-1}\overline{Z}_{l-1}. \tag{C.6}$$

Substituting (C.6) into (C.4) and (C.5), the necessary conditions become:

- Bias Vector: $\widehat{b}_{1l} = \overline{b}_{1l}$,

- Query and Key: $(W_{Kl}^h + \Delta W_{Kl}^h)^\top(W_{Ql}^h + \Delta W_{Ql}^h) = W_{2,l-1}^{-1\top}\overline{W}_{2,l-1}^\top\overline{W}_{Kl}^{h\top}\overline{W}_{Ql}^h\overline{W}_{2,l-1}W_{2,l-1}^{-1}$,

- Value and Output Projection: $(W_{Ol}^h + \Delta W_{Ol}^h)(W_{Vl}^h + \Delta W_{Vl}^h) = W_{1l}^{-1}\overline{W}_{1l}\overline{W}_{Ol}^h\overline{W}_{Vl}^h\overline{W}_{2,l-1}W_{2,l-1}^{-1}$.

By setting $\widehat{b}_{1l} = \overline{b}_{1l}$ and adjusting $\Delta W_{Kl}^h, \Delta W_{Ql}^h, \Delta W_{Ol}^h, \Delta W_{Vl}^h$ for all $h \in [H]$ based on Lemma C.1, we satisfy all three conditions above, thereby obtaining $\widehat{H}_l = \overline{H}_l$ for $l \in [L]\backslash\{1\}$.

**Output Layer Analysis.** By applying the induction method, we have established $\widehat{H}_l = \overline{H}_l$ for all $l \in [L]$. We only need to select appropriate weight matrices to ensure that $\overline{f}(X) = f(X)$ for all $X \in \mathcal{X}$. The final output of the target model $\overline{f}$ with input $X$ can be written as

$$\overline{f}(X) = \text{Softmax}_1(\overline{W}_o\overline{Z}_L) = \text{Softmax}_1(\overline{W}_o(\overline{W}_{2L}\overline{H}_L + \overline{b}_{2L}\mathbf{1}_N^\top)).$$

Similarly, the final output of the adapted model $f$ with input $X$ can be written as

$$f(X) = \text{Softmax}_1((W_o + \Delta W_o)\widehat{Z}_L)$$

$$= \text{Softmax}_1((W_o + \Delta W_o)((W_{2L}\Delta W_{2L})\widehat{H}_L + \widehat{b}_{2L}\mathbf{1}_N^\top)).$$

To achieve $\bar{f}(X) = f(X)$, we select $\Delta W_{2L}$ and $\Delta W_o$ based on Lemma C.1, and let $\widehat{b}_{2L} = (W_o + \Delta W_o)^{-1}\overline{W}_o\bar{b}_{2L}$, where $W_o + \Delta W_o$ is invertible as shown in the proof of Lemma C.1. Combining above, we complete the proof. $\square$

## D  POST-TRAINING QUANTIZATION EVALUATION

We evaluated our methods across OPT-125m, OPT-350m, and OPT-1.3b using three PTQ methods: SmoothQuant (Xiao et al., 2023), AffineQuant (Ma et al., 2024), and OmniQuant (Shao et al., 2023). Performance was assessed using Text Perplexity (PPL), SpeechLM PPL, Word Error Rate (WER) for Automatic Speech Recognition (ASR), and WER for Text-to-Speech (TTS). Detailed results for W8A8 and W4A4 configurations are presented in Table 6.

Table 6: **Comparing SpARQ with Vanilla Framework in a Post-Training Quantization (PTQ) setting.** Experiments were conducted across three quantization methods (SmoothQuant, AffineQuant, omniQuant) and two weight and activation quantization configurations (W8A8, W4A4). Evaluation metrics included Text PPL, SpeechLM PPL, ASR WER, and TTS WER. We assessed the average performance drop across these four tasks post-quantization. Results show SpARQ consistently outperforms Vanilla framework, exhibiting smaller performance drops in most configurations, demonstrating its superior efficiency in PTQ settings.

| Model | Method | #Bits | Quantization Method | TextLM PPL ($\downarrow$) | SpeechLM PPL ($\downarrow$) | ASR WER ($\downarrow$) | TTS WER ($\downarrow$) | Avg Performance Drop ($\downarrow$) |
|---|---|---|---|---|---|---|---|---|
| OPT-125m | Vanilla | W16/A16 | - | 22.56 | 59.42 | 12.40 | 12.08 | - |
| | | W8/A8 | SmoothQuant | 22.63 | 59.53 | 12.46 | 12.38 | 0.85% |
| | | W8/A8 | AffineQuant | 22.61 | 59.52 | 12.42 | 12.37 | 0.74% |
| | | W8/A8 | omniQuant | 22.62 | 59.53 | 12.44 | 12.38 | 0.81% |
| | | W4/A4 | SmoothQuant | 45.23 | 96.87 | 52.31 | 48.79 | 197.31% |
| | | W4/A4 | AffineQuant | 31.25 | 80.19 | 29.44 | 28.34 | 86.37% |
| | | W4/A4 | omniQuant | 31.28 | 80.21 | 31.98 | 29.55 | 94.04% |
| | SpARQ | W16/A16 | - | 22.70 | 59.45 | 12.61 | 12.11 | - |
| | | W8/A8 | SmoothQuant | 22.71 | 59.49 | 12.64 | 12.18 | **0.23%** |
| | | W8/A8 | AffineQuant | 22.71 | 59.48 | 12.62 | 12.13 | **0.08%** |
| | | W8/A8 | omniQuant | 22.71 | 59.49 | 12.63 | 12.14 | **0.13%** |
| | | W4/A4 | SmoothQuant | 37.14 | 84.55 | 35.32 | 36.73 | **112.05%** |
| | | W4/A4 | AffineQuant | 26.11 | 68.42 | 14.33 | 15.71 | **18.52%** |
| | | W4/A4 | omniQuant | 26.12 | 68.63 | 14.53 | 16.01 | **19.63%** |
| OPT-350m | Vanilla | W16/A16 | - | 13.13 | 43.10 | 8.42 | 17.56 | - |
| | | W8/A8 | SmoothQuant | 13.17 | 43.14 | 8.47 | 17.71 | 0.46% |
| | | W8/A8 | AffineQuant | 13.15 | 43.12 | 8.45 | 17.66 | 0.28% |
| | | W8/A8 | omniQuant | 13.15 | 43.12 | 8.46 | 17.68 | 0.33% |
| | | W4/A4 | SmoothQuant | 36.74 | 75.38 | 40.17 | 70.53 | 233.37% |
| | | W4/A4 | AffineQuant | 27.28 | 66.31 | 36.84 | 40.83 | 157.92% |
| | | W4/A4 | OmmiQuant | 27.85 | 67.83 | 37.54 | 41.37 | 162.73% |
| | SpARQ | 16/16A | - | 13.47 | 43.34 | 9.81 | 17.31 | - |
| | | W8/A8 | SmoothQuant | 13.50 | 43.39 | 9.88 | 17.38 | **0.36%** |
| | | W8/A8 | AffineQuant | 13.48 | 43.36 | 9.85 | 17.35 | **0.19%** |
| | | W8/A8 | omniQuant | 13.48 | 43.36 | 9.85 | 17.37 | **0.22%** |
| | | W4/A4 | SmoothQuant | 23.48 | 62.17 | 36.22 | 40.83 | **130.71%** |
| | | W4/A4 | AffineQuant | 22.82 | 51.74 | 25.78 | 28.44 | **78.97%** |
| | | W4/A4 | OmmiQuant | 22.83 | 52.08 | 26.11 | 29.15 | **81.05%** |
| OPT-1.3b | Vanilla | W16/A16 | - | 12.62 | 41.33 | 8.00 | 18.73 | - |
| | | W8/A8 | SmoothQuant | 12.68 | 41.48 | 8.14 | 18.80 | 0.74% |
| | | W8/A8 | AffineQuant | 12.65 | 41.46 | 8.12 | 18.75 | 0.54% |
| | | W8/A8 | omniQuant | 12.66 | 41.46 | 8.12 | 18.75 | 0.56% |
| | | W4/A4 | SmoothQuant | 36.74 | 87.46 | 48.96 | 53.15 | 249.63% |
| | | W4/A4 | AffineQuant | 24.31 | 61.74 | 43.68 | 32.47 | 165.33% |
| | | W4/A4 | OmmiQuant | 24.43 | 62.38 | 44.52 | 33.03 | 169.34% |
| | SpARQ | 16/16A | - | 12.95 | 42.48 | 8.25 | 12.07 | - |
| | | W8/A8 | SmoothQuant | 13.00 | 42.49 | 8.33 | 12.11 | **0.43%** |
| | | W8/A8 | AffineQuant | 12.96 | 42.48 | 8.29 | 12.08 | **0.16%** |
| | | W8/A8 | omniQuant | 12.98 | 42.48 | 8.31 | 12.10 | **0.30%** |
| | | W4/A4 | SmoothQuant | 23.83 | 58.33 | 32.27 | 33.12 | **146.72%** |
| | | W4/A4 | AffineQuant | 20.81 | 48.84 | 22.78 | 25.46 | **90.68%** |
| | | W4/A4 | OmmiQuant | 20.88 | 48.97 | 23.58 | 26.83 | **96.15%** |

# E  HYPER-PARAMETER SETTINGS

## E.1  LoRA TRAINING/FINE-TUNING

We present the hyperparameters used in the low-rank adaptation stage for each model. We use **Adam** (Kingma, 2014) as the optimizer. Most of the other hyperparameters remain the same across all models and datasets, including a batch bin size of 5000, a warmup step of 2500, and a weight decay of $1e^{-6}$. A learning rate of $3e^{-4}$ is used for all models during fine-tuning. For low-rank adaptation, we apply a dropout rate of 0.05, with a LoRA rank and alpha both set to 256. We fine-tune the attention module weights $W_k$, $W_q$, $W_v$, $W_o$, along with the MLP layer in the attention mechanism.

## E.2  POST-TRAINING QUANTIZATION

For SmoothQuant, we adopted the recommended hyperparameter $\alpha$=0.5, as suggested in the original paper. This value typically provides a balanced trade-off between smoothing activations and adjusting weights. In the case of OmniQuant, we set the group size to 128. This parameter determines the granularity of the quantization process, balancing between quantization accuracy and computational efficiency. For AffineQuant, we configured the stability factor to 0.01. This factor helps in maintaining numerical stability during the quantization process, particularly for values close to zero. Across all three PTQ methods, we utilized a consistent calibration batch size of 256.

## E.3  HIFI-GAN DECODER FOR SPEECH SYNTHESIS

Within our SpeechLM framework, the TTS system synthesizes speech by converting the discrete speech tokens generated by SpeechLM into speech waveforms using a HiFi-GAN-based vocoder. These discrete speech tokens are derived from a pre-trained HuBERT model with a dictionary size of 200 (HuBERT k=200). The HiFi-GAN vocoder is trained on the LJSpeech-1.1 dataset, downsampled to 16 kHz, using the HuBERT k=200 unit representations as input features.

The vocoder configuration includes 200 discrete unit embeddings, each with a dimensionality of 128. The model utilizes a ResBlock type 1 architecture with upsampling rates of [5, 4, 4, 2, 2] and corresponding kernel sizes of [11, 8, 8, 4, 4]. The initial channel size is set to 512, and the vocoder operates at a 16 kHz sampling rate. We use the Adam optimizer with an initial learning rate of 0.0008. Training is performed with a batch size of 64 and a code hop size of 320 to ensure proper alignment between discrete units and waveform segments.

To evaluate the quality of the synthesized speech from our TTS system, we employ an ASR system based on OpenAI's Whisper model. The synthesized speech is fed into Whisper to obtain transcriptions, which are then compared to the ground truth text to compute the Word Error Rate (WER). This WER serves as a measure of the quality of the synthesized speech within the SpeechLM framework.

# F  OUTLIER EVALUATION SETTINGS

We report the maximum infinity norm $\|\mathbf{x}\|_\infty$ of activation tensors $\mathbf{x}$ across all Transformer layers as a metric to detect outliers. We also report the average kurtosis of $\mathbf{x}$, computed by averaging across all output components in the Transformer layers. Those two metrics have been demonstrated to strongly correlate with the model's ability (i.e., robustness against outliers) to be quantized, as shown in previous studies (Bondarenko et al., 2021; Hu et al., 2024a).

# G  ADDTIONAL EXPERIMENT

## G.1  COMPARISON WITH CLIPPED SOFTMAX AND GATED ATTENTION

| Model | Method | W/A Bits | Text PPL ↓ | Speech PPL ↓ | ASR WER ↓ | TTS WER ↓ | Avg Performance Drop Rate |
|---|---|---|---|---|---|---|---|
| OPT-350m | Vanilla | 8/8 | 13.17 | 43.14 | 8.47 | 17.71 | 0.46% |
| | SpARQ | 8/8 | 13.50 | 43.39 | 9.88 | 17.38 | **0.36%** |
| | Clipped Softmax | 8/8 | 13.16 | 43.16 | 8.47 | 17.71 | 0.45% |
| | Gated Attention | 8/8 | 13.16 | 43.15 | 8.47 | 17.70 | 0.43% |
| | Vanilla | 4/4 | 36.74 | 75.38 | 40.17 | 70.53 | 233.36% |
| | SpARQ | 4/4 | 23.48 | 62.17 | 36.22 | 40.83 | **130.71%** |
| | Clipped Softmax | 4/4 | 35.86 | 73.91 | 38.82 | 68.35 | 223.72% |
| | Gated Attention | 4/4 | 35.26 | 73.02 | 37.44 | 66.29 | 215.03% |

We have conducted additional experiments comparing SpARQ with these alternative techniques. The results demonstrate that at 8-bit quantization, all methods perform similarly well, with performance drops under 0.5%. This suggests that for moderate quantization, the choice of outlier mitigation strategy is less critical. However, the differences become pronounced at 4-bit quantization, where SpARQ achieves a 130.71% performance drop compared to 215.03% for Gated Attention, 223.72% for Clipped Softmax, and 233.36% for the vanilla approach. SpARQ's superior performance under aggressive quantization (4-bit) can be attributed to its architectural approach to outlier mitigation, which addresses the issue at its source rather than merely limiting attention values. This is particularly important for speech-text cross-modal tasks where maintaining precise attention patterns is crucial for accurate modality fusion. These results validate SpARQ's effectiveness compared to alternative attention modification approaches, particularly in challenging low-bit quantization scenarios.

### G.2 STABILIZATION MODULE TO A STANDARD TRANSFORMER

We have conducted an additional experiment to isolate the effect of max-shift stabilization by applying it to a standard Transformer architecture. The results show that applying max-shift stabilization alone to a standard Transformer provides minimal improvement (34.4% vs 34.3%

Table 7: Vanilla Transformer with Stabilization.

| Model | Stabilized Method | Val Accuracy (%) |
|---|---|---|
| Vanilla OPT-1.3b | N/A | 34.3 |
| | Max-Shift | **34.4** |

validation accuracy). This suggests that the significant performance improvements we observe in SpARQ (as shown in Tables 1 and 2) are primarily attributable to the synergistic combination of the outlier-free Hopfield layer and max-shift stabilization, rather than either component alone. The minimal improvement from stabilization alone is understandable given that max-shift stabilization was specifically designed to address numerical stability issues that arise when using the outlier-free layer, rather than to directly improve model performance. These results help clarify that SpARQ's effectiveness comes from the complementary nature of its components: the outlier-free layer provides the fundamental mechanism for outlier mitigation, while max-shift stabilization enables practical training with this architecture.

### G.3 APPLYING PTQ TO LoRA-TRAINED MODELS

To provide a complete picture of how SpARQ performs when combining these techniques, we conducted additional experiments applying post-training quantization to LoRA-trained models. The results demonstrate that SpARQ maintains its advantages even when combining LoRA with post-training quantization (SQ). When applying 8-bit quantization to LoRA-trained models, SpARQ exhibits a significantly lower performance drop (155.95%) compared to the vanilla framework (396.34%). This advantage persists even under more aggressive 4-bit quantization, where SpARQ shows a 186.66% performance drop versus 417.52% for the vanilla framework. Most notably, SpARQ's ASR performance remains relatively stable under quantization (18.52% to 28.32% WER), while the vanilla framework shows severe degradation (93.91% to 97.24% WER). These results further validate SpARQ's effectiveness in handling outliers, as it maintains better performance not just in individual scenarios (as shown in Tables 1 and 2) but also when combining low-rank adaptation with quantization. This is particularly important for practical applications where both model compression techniques might need to be applied simultaneously.

Table 8: Results of combining LoRA with Post-Training Quantization (PTQ) on OPT-350m. We evaluate the performance when applying Smooth Quantization (SQ) to LoRA-trained models, OPT-350m, for both vanilla and SpARQ frameworks. The results show SpARQ maintains better performance when combining these compression techniques.

| Model | Method | W/A Bits | Text PPL ↓ | Speech PPL ↓ | ASR WER ↓ | TTS WER ↓ | Avg Performance Drop Rate |
|---|---|---|---|---|---|---|---|
| | LoRA | 16/16 | 17.87 | 51.65 | 93.91 | 97.06 | 381.00% |
| Vanilla | LoRA + SQ | 8/8 | 20.54 | 56.88 | 95.36 | 99.11 | 396.34% |
| | LoRA + SQ | 4/4 | 27.31 | 60.03 | 97.24 | 99.73 | 417.52% |
| | LoRA | 16/16 | 17.27 | 50.14 | 18.52 | 75.18 | 116.75% |
| SpARQ | LoRA + SQ | 8/8 | 18.91 | 53.24 | 25.47 | 86.71 | 155.95% |
| | LoRA + SQ | 4/4 | 25.89 | 59.11 | 28.32 | 91.63 | 186.66% |

## G.4 WEIGHT CLIPPING DURING PTQ

Weight clipping is an important baseline to include. The experiments use percentile-based clipping with smoothquant (clip 1% tail weights and 5% tail weights) show as below:

Table 9: Comparison of quantization results with different weight clipping strategies. Results show that traditional weight clipping leads to performance degradation in multimodal settings, while SpARQ performs best without clipping.

| Method | W/A | Text PPL ↓ | Speech PPL ↓ | ASR WER ↓ | TTS WER ↓ | Avg Performance Drop Rate ↓ |
|---|---|---|---|---|---|---|
| Vanilla + SQ | 8/8 | 13.17 | 43.14 | 8.47 | 17.71 | 0.46% |
| Vanilla+SQ+Clip-1% | 8/8 | 21.33 | 52.53 | 13.71 | 26.26 | 49.18% |
| Vanilla+SQ+Clip-5% | 8/8 | 25.34 | 55.03 | 16.83 | 28.91 | 71.30% |
| SpARQ+SQ | 8/8 | 13.50 | 43.39 | 9.88 | 17.38 | **0.36%** |
| SpARQ+SQ+Clip-1% | 8/8 | 21.14 | 50.87 | 13.31 | 25.42 | **39.21%** |
| SpARQ+SQ+Clip-5% | 8/8 | 24.37 | 52.77 | 16.22 | 27.04 | **55.75%** |
| Vanilla + SQ | 4/4 | 36.74 | 75.38 | 40.17 | 70.53 | 233.36% |
| Vanilla+SQ+Clip-1% | 4/4 | 36.22 | 78.33 | 40.46 | 75.88 | 267.56% |
| Vanilla+SQ+Clip-5% | 4/4 | 40.73 | 82.51 | 45.14 | 80.02 | 273.36% |
| SpARQ+SQ | 4/4 | 23.48 | 62.17 | 36.22 | 40.83 | **130.71%** |
| SpARQ+SQ+Clip-1% | 4/4 | 35.11 | 70.21 | 37.29 | 68.08 | **199.02%** |
| SpARQ+SQ+Clip-5% | 4/4 | 37.03 | 74.24 | 39.16 | 70.31 | **212.89%** |

## G.5 QLoRA WITH LOWER BITS

We investigate how different quantization levels affect LoRA-adapted models. The results show that SpARQ maintains its benefits under aggressive quantization, reducing performance degradation from 330.47% to 136.07% at 8-bit quantization and from 387.50% to 176.52% at 4-bit quantization. The performance degradation with SpARQ shows a more gradual pattern across quantization levels, suggesting better stability under increasingly aggressive compression. Furthermore, the confounding effects between LoRA and quantization are significantly mitigated by SpARQ, as evidenced by the smaller performance gaps between 8-bit and 4-bit configurations.

Table 10: Comparison of QLoRA performance under different quantization settings. Results demonstrate SpARQ's superior ability to maintain performance under aggressive quantization compared to the vanilla framework.

| Method | W/A | Text PPL ↓ | Speech PPL ↓ | ASR WER ↓ | TTS WER ↓ | Avg Performance Drop Rate ↓ |
|---|---|---|---|---|---|---|
| Vanilla+QLoRA | 8/8 | 18.07 | 51.50 | 76.04 | 98.63 | 330.47% |
| Vanilla+QLoRA | 4/4 | 25.83 | 58.24 | 88.71 | 99.14 | 387.50% |
| SpARQ+QLoRA | 8/8 | 16.64 | 48.34 | 22.33 | 83.36 | 136.07% |
| SpARQ+QLoRA | 4/4 | 23.45 | 56.33 | 27.36 | 90.55 | 176.52% |

## H LORA PERCENTAGE

We conducted an analysis of Low-Rank Adaptation (LoRA) parameters across different sizes of Open Pre-trained Transformer (OPT) models. Our findings are summarized in the table below, which compares the number of LoRA parameters to the full model parameters for three OPT variants.

Table 11: LoRA Parameters Comparison for OPT Models

| Model | LoRA Parameters | Full Model Parameters | LoRA Percentage |
|---|---|---|---|
| OPT-125M | 9.4M | 125M | 7.5% |
| OPT-350M | 25.2M | 350M | 7.2% |
| OPT-1.3B | 50.3M | 1.3B | 3.9% |

