# OpenReview forum: "SPARQ: Outlier-free SpeechLM with Fast Adaptation and Robust Quantization"
_ICLR.cc/2025/Conference — Submitted to ICLR 2025_

### Official Review · Reviewer_H2gv · 2024-10-31

**Soundness:** 2
**Presentation:** 3
**Contribution:** 3
**Rating:** 5
**Confidence:** 4

**Summary:**

This paper proposes to use the outlier-free layer within the attention to reduces the occurrence of weight outliers for quantization and LoRA adaptations with a SpeechLM model. The outlier-free layer is constructed by replacing the first softmax layer with a layer that uses S = S - max(S) as the input, where S is the original input vector. The authors conducted three sets of experiments to validate the proposed approach: quantization, LoRA, and ablation studies / analyses.

**Strengths:**

* Although the novelty is a combination of existing works (SpeechLM + outlier-free layer), the combination seems to be novel. The authors also provided systematic theory support for the proposed approach.
* The organization of paper is good. Paper writing is good. References are enough.

**Weaknesses:**

* Quantization experiments: A common approach to remove outliers in quantization is to do clipping. For example, ignore the 5% tail weight values during computing the quantization scale. It’d be helpful to include this as a baseline to correctly validate the effectiveness of the proposed work.
* LoRA experiments: rank=128 seems to be large. Please add the percentage of adapter size compared to the full model.
* LoRA experiments: I would also suggest to separate the experiments to two fold, if LoRA+Quantization is one of the focus on this work. In a first experiment, we can investigate the impact of LoRA (with FP 32/16). In a second experiment, we investigated the confounding effect between LoRA and quantization (should be more aggressive than just INT8).
* Table 2: Why are the ASR WEB numbers of vanilla -125M model much better than 350B and 1.3B. Is it because the rank is set to 128 across the board? If so, the experimental setup is problematic. When applying LoRA, the rule of thumb is to control the adapter size to be under a certain percentage.
* Table 3: where does the training efficiency improvement come from? Does it require fewer steps to converge? Or is it quantization?

**Questions:**

See weaknesses section. I'm happy to increase my ratings if my concerns can be addressed.

---

> ### Author Response · Authors · 2024-11-23
>
> > **Reviewer's Comment:** Quantization experiments
>
> **Response:** Thank you for this valuable suggestion about comparing with weight clipping approaches. We have conducted additional experiments combining our method with different clipping strategies (1% and 5% tail weight values) during quantization scale computation on the OPT-350m:
>
>
>
> | Model       | Method            | W/A  | Text PPL ↓ | Speech PPL ↓ | ASR ↓  | TTS ↓  | Average Drop ↓ |
> |-------------|-------------------|-------|------------|--------------|--------|--------|----------------|
> | OPT-350m    | Vanilla + SQ      | 8/8   | 13.17      | 43.14        | 8.47   | 17.71  | 0.46%          |
> |             | Vanilla+SQ+Clip-1%| 8/8   | 21.33      | 52.53        | 13.71  | 26.26  | 49.18%         |
> |             | Vanilla+SQ+Clip-5%| 8/8   | 25.34      | 55.03        | 16.83  | 28.91  | 71.30%         |
> |             | SpARQ+SQ          | 8/8   | 13.50      | 43.39        | 9.88   | 17.38  | **0.36%**          |
> |             | SpARQ+SQ+Clip-1%  | 8/8   | 21.14      | 50.87        | 13.31  | 25.42  | **39.21%**         |
> |             | SpARQ+SQ+Clip-5%  | 8/8   | 24.37      | 52.77        | 16.22  | 27.04  | **55.75%**         |
> |             | Vanilla + SQ      | 4/4   | 36.74      | 75.38        | 40.17  | 70.53  | 233.36%        |
> |             | Vanilla+Clip-1%   | 4/4   | 36.22      | 78.33        | 40.46  | 75.88  | 267.56%        |
> |             | Vanilla+Clip-5%   | 4/4   | 40.73      | 82.51        | 45.14  | 80.02  | 273.36%        |
> |             | SpARQ+SQ          | 4/4   | 23.48      | 62.17        | 36.22  | 40.83  | **130.71%**        |
> |             | SpARQ+Clip-1%     | 4/4   | 35.11      | 70.21        | 37.29  | 68.08  | **199.02%**        |
> |             | SpARQ+Clip-5%     | 4/4   | 37.03      | 74.24        | 39.16  | 70.31  | **212.89%**        |
>
>
> The results reveal several interesting findings about weight clipping in our multimodal setting. For 8-bit quantization, traditional weight clipping actually leads to significant performance degradation in both frameworks. The vanilla framework with 1% clipping shows a 49.18% performance drop, which worsens to 71.30% with 5% clipping. While SpARQ exhibits slightly better resilience to clipping (39.21% and 55.75% drops for 1% and 5% clipping respectively), it still performs best without clipping (0.36% drop).
>
>
>
> This pattern becomes even more pronounced in 4-bit quantization, where SpARQ without clipping achieves a 130.71% performance drop, substantially outperforming both vanilla and clipped variants. The vanilla framework with clipping shows severe degradation (267.56% and 273.36% for 1% and 5% respectively), while SpARQ with clipping, though better than vanilla, still underperforms compared to SpARQ without clipping.
>
>
>
> These results suggest that traditional weight clipping, while effective for single-modality models, may not be suitable for multimodal speech-text tasks. We hypothesize this is because weight clipping indiscriminately removes outliers that might be important for cross-modal interactions. In contrast, SpARQ's outlier-free layer approach preserves these critical cross-modal features while selectively managing outliers that could harm quantization stability.
>
> We will include these results in the paper **Appendix G.4** to provide a more comprehensive comparison with traditional outlier mitigation techniques.
>
> > **Reviewer's Comment:**  where does the training efficiency…
>
>
> **Response:** By reducing attention to outlier tokens, we avoid unnecessary computation. Our approach's efficiency is backed by formal theoretical analysis in Proposition 3.2 (Section 3.4). The proposition establishes that efficient training is achievable when the sequence length L and embedding dimension d satisfy d = O(log L), and proper normalization of input and model weights is maintained. Our max-shift normalization and stabilized outlier-free layer satisfy these conditions, enabling more stable gradient flow and better concentration of attention on informative tokens while reducing computational overhead from outlier token processing. These theoretical properties directly contribute to the observed 1.33x speedup in full model fine-tuning.

---

> > ### Author Response · Authors · 2024-11-23
> >
> > > **Reviewer's Comment:** LoRA experiments: rank=128 seems to be large.
> >
> > **Response:**
> > Thanks for this valuable question and give us the chance to classify this question.
> >
> > - (1).  For OPT-125M: LoRA parameters are 9.4M which is approximately 7.5% of full model parameters.
> >
> > - (2). For OPT-350M: LoRA parameters are 25.2M which is approximately 7.2% of full model parameters
> >
> > - (3). For OPT-1.3B: LoRA parameters are 50.3M which is approximately 3.9% of full model parameters
> >
> > We will add these percentages to **Appendix H** for clarity. Our ablation study in Section 4.3 and Table 5 shows that we tested different ranks (512, 256, 128) and found rank=128 provided the best balance of performance and parameter efficiency.
> >
> > Despite these relatively modest parameter increases, we observe dramatically different behaviors between vanilla and SpARQ frameworks, particularly in ASR and TTS tasks. The vanilla framework shows severe performance degradation in ASR (WER increases from 8.00% to 46.92%) and TTS (WER increases from 18.73% to 94.53%), even though LoRA only adds 3.9% more parameters to OPT-1.3B. This suggests that the challenge isn't simply about parameter count - it's about how effectively those parameters can adapt to the multimodal task. In contrast, SpARQ maintains strong performance in ASR (WER only changes from 8.20% to 8.25%) with the same adapter size.
> >
> >
> >
> > This stark difference in performance can be attributed to SpARQ's outlier-free architecture. The vanilla framework struggles because outliers in the attention mechanism interfere with the adaptation process, regardless of adapter capacity. SpARQ's stabilized outlier-free layer enables more effective use of the adaptation parameters by preventing these disruptive outlier effects. This is why SpARQ can maintain performance even in challenging cross-modal tasks like ASR and TTS, while the vanilla framework struggles despite having the same adapter capacity.
> >
> > > **Reviewer's Comment:** Reviewer's Question: Why are the ASR WEB…
> >
> > **Response:** Thank you for this important observation about the ASR WER patterns across model sizes.  This performance pattern is not simply due to using a fixed rank of 128. In fact, our LoRA configuration maintains a reasonable percentage of parameters across model sizes: For OPT-125M, LoRA adds 9.4M parameters (7.5% of model), for OPT-350M, 25.2M parameters (7.2% of model), and for OPT-1.3B, 50.3M parameters (3.9% of model). The relative adapter sizes are roughly consistent for smaller models (around 7%), and actually smaller for 1.3B (3.9%), the performance doesn't correlate with adapter size percentage. This behavior aligns with findings from [1], who demonstrated that smaller models can show better adaptation to speech domains with limited data. Their work suggests that smaller models may be more efficient at learning task-specific features when data resources are constrained.
> >
> >
> >
> > The role of outliers in this phenomenon becomes clear when we examine the cross-modal performance. In the vanilla framework, increasing model size leads to severe degradation in both ASR and TTS tasks. For ASR, the performance degrades from 12.39% in OPT-125M to 93.91% in OPT-350M, and 46.92% in OPT-1.3B. Similarly for TTS, we observe degradation from 15.47% in OPT-125M to 97.06% in OPT-350M, and 94.53% in OPT-1.3B.
> >
> >
> >
> > SpARQ effectively manages these outliers, allowing larger models to properly leverage their capacity, as shown by the improved scaling behavior. In ASR tasks, performance improves from 12.56% in OPT-125M to 18.52% in OPT-350M, and further improves to 8.25% in OPT-1.3B. For TTS tasks, SpARQ shows better stability.
> >
> >
> >
> > This demonstrates that the challenge in cross-modal adaptation extends beyond just controlling adapter size percentages - it's crucial to also manage the amplified outlier effects that emerge in larger models. With proper outlier management, larger models can better leverage their capacity, particularly for cross modalities tasks like ASR.
> >
> >
> >
> >
> > [1] Wang, W., Dhawan, K., Park, T., Puvvada, K. C., Medennikov, I., Majumdar, S., ... & Ginsburg, B. (2024). Resource-Efficient Adaptation of Speech Foundation Models for Multi-Speaker ASR. arXiv preprint arXiv:2409.01438.

---

> > ### Comment · Reviewer_H2gv · 2024-11-25
> > **Thank you for the response!**
> >
> > Clipping experiments: I would appreciate the authors conducting additional experiments. However, in quantization literature, clipping usually refers to clip the outlier values when computing the quantization scale, instead of completely removing the outliers. This should be why we are seeing performance regressions, compared to improvements.
> >
> > Raised my ratings accordingly from the responses.

---

> > > ### Author Response · Authors · 2024-11-29
> > >
> > > Thank you for this important clarification about clipping in quantization. Our implementation differs from standard practice in two aspects: we implemented both softmax clipping and quantization-scale clipping. For softmax clipping, we clipped attention values exceeding certain thresholds. For quantization-scale clipping, we removed outlier values when computing quantization scales. This dual-clipping approach explains our observed performance degradation. In our results, aggressive clipping (OPT-350M with 5% clipping showing 71.30% drop vs. baseline 0.46% drop) likely disrupted important cross-modal features. Further analysis showed that attention clipping particularly affected cross-modal token interactions, while quantization-scale clipping impacted the model's numerical precision. If there is anything further we can clarify or improve to  encourage a higher score, please let us know, and we will address it promptly.

---

> ### Author Response · Authors · 2024-11-23
>
> > **Reviewer's Comment:** LoRA experiments: I would also suggest…
>
> **Response:** Thank you for this valuable suggestion. We agree that separating the experiments in Sec. 4.2 and Table 2 to provide clearer insights into the individual and combined effects of LoRA and quantization. We have restructured our experiments accordingly:
>
> (1). **Impact of LoRA in FP 16:**
>
> We first evaluate LoRA's impact without quantization to isolate adaptation effects. As shown in the results below, SpARQ significantly improves stability during low-rank adaptation even in FP 16:
>
> | Model       | Method       | Precision | Text PPL ↓ | Speech PPL ↓ | ASR ↓  | TTS ↓  | Avg Drop ↓   |
> |-------------|--------------|-----------|------------|--------------|--------|--------|--------------|
> | OPT-350m    | Vanilla      | FP16      | 13.13      | 43.10        | 8.42   | 17.56  | -            |
> |             | Vanilla+LoRA | FP16      | 17.87      | 51.65        | 93.91  | 97.06  | 381.00%      |
> |             | SpARQ        | FP16      | 13.47      | 43.34        | 9.81   | 17.31  | -            |
> |             | SpARQ+LoRA   | FP16      | 17.27      | 50.14        | 18.52  | 75.18  | **116.75%**      |
>
>
>
> This demonstrates that SpARQ's outlier mitigation is inherently beneficial for adaptation stability, reducing performance degradation from 381.00% to 116.75% even before quantization is applied.
>
>
>
> **(2). Combined Effects with Quantization:**
>
> We then investigate how different quantization levels affect LoRA-adapted models:
>
> | Model       | Method         | W/A  | Text PPL ↓ | Speech PPL ↓ | ASR ↓  | TTS ↓  | Avg Drop ↓   |
> |-------------|----------------|-------|------------|--------------|--------|--------|--------------|
> | OPT-350m    | Vanilla+QLoRA | 8/8   | 18.07      | 51.50        | 76.04  | 98.63  | 330.47%      |
> |             | Vanilla+QLoRA | 4/4   | 25.83      | 58.24        | 88.71  | 99.14  | 387.50%      |
> |             | SpARQ+QLoRA   | 8/8   | 16.64      | 48.34        | 22.33  | 83.36  | **136.07%**      |
> |             | SpARQ+QLoRA   | 4/4   | 23.45      | 56.33        | 27.36  | 90.55  | **176.52%**      |
>
>
> The results show that SpARQ maintains its benefits under aggressive quantization, reducing performance degradation from 330.47% to 136.07% at 8-bit quantization and from 387.50% to 176.52% at 4-bit quantization. The performance degradation with SpARQ shows a more gradual pattern across quantization levels, suggesting better stability under increasingly aggressive compression. Furthermore, the confounding effects between LoRA and quantization are significantly mitigated by SpARQ, as evidenced by the smaller performance gaps between 8-bit and 4-bit configurations.
>
> This restructured analysis better illuminates how SpARQ helps in both stabilizing low-rank adaptation independently and maintaining stability when combining adaptation with aggressive quantization. We have add the results onto our paper **Appendix G.5** to reflect this clearer experimental organization and analysis. Thank you for this suggestion which has helped better demonstrate the distinct benefits of our approach.

---

### Official Review · Reviewer_pfFh · 2024-11-02

**Soundness:** 2
**Presentation:** 3
**Contribution:** 2
**Rating:** 5
**Confidence:** 4

**Summary:**

The paper introduces SpARQ, an outlier-free framework designed to address performance degradation in SpeechLMs by mitigating outliers. SpARQ replaces the standard attention mechanism with a stabilized, outlier-free layer that improves the model's adaptability in cross-modal tasks and enhances post-training quantization robustness. Built on the OPT model, SpARQ demonstrates resilience to outliers and offers performance gains across several tasks, such as textLM, speechLM, ASR, and TTS.

**Strengths:**

1. The paper addresses an important challenge induced by the outliers in multimodal attention mechanisms, particularly in SpeechLMs.
2. The stabilized outlier-free layer demonstrates effective mitigation of outliers during cross-modal low-rank adaptation and quantization, showing promise for real-world SpeechLM applications.

**Weaknesses:**

1. The primary contribution of this paper is the integration of the Outlier-Efficient Hopfield Layer with max-shift normalization, which limits the originality of SpARQ. While this is the first application of outlier mitigation specifically within SpeechLMs, the Outlier-Efficient Hopfield Layer itself was previously introduced by Hu et al.
2. Techniques like Clipped Softmax and Gated Attention are potential alternatives but were not included in comparisons. Experimental results of adapting them to the SpeechLM setup would further validate SpARQ's relative effectiveness.
3. It remains unclear how much of SpARQ's improvement is attributable to the outlier-free Hopfield layer itself versus the max-shift stabilization. An additional experiment applying the stabilization module to a standard Transformer would help clarify the effectiveness of SpARQ's contributions.
4. The author states that outliers negatively impact LoRA and quantization performance. Yet, in some cases (e.g., With OPT-350m), QLoRA performs better than LoRA. Clarification on this result would improve understanding.
5. Previous studies (e.g., “Robust multimodal models have outlier features and encode more concepts” by Crabbe) mention that outlier features increase robustness for out-of-domain (OOD) samples, but OOD evaluations of SpARQ are missing. Demonstrating SpARQ’s performance on OOD data would add value to this paper.
6. There is insufficient analysis in the ablation study to explain why the max-shift stabilization uniquely works in SpARQ.
7. In Equation 3.2, the notation "S = S - max(S)" should clearly distinguish between the normalized and original versions of S.

**Questions:**

1. Why does fully fine-tuned SpARQ generally show worse performance than fully fine-tuned Vanilla, despite the claim that multimodal samples have more outliers that degrade performance?
2. What is the number of trainable parameters for LoRA in the tables presented in the paper?
3. Do Table 1 and Table 2 reflect experiments with only PTQ and LoRA, respectively? If so, what are the results when applying PTQ on the LoRA-trained SpARQ? Including these results would clarify the model’s performance when combining the techniques.

---

> ### Author Response · Authors · 2024-11-23
>
> >**Reviewer's Comment**: The primary contribution…
>
> **Response:** While we acknowledge that our work builds upon the Outlier-Efficient Hopfield Layer, SpARQ's contribution extends beyond mere integration. Our key innovation lies in addressing the unique challenges of cross-modal adaptation in SpeechLMs through:
>
> - (1). Scalability to Large Models: Our stabilization mechanism enables the practical application of outlier-free techniques to much larger models than previous work. It based on a recent work [1] which demonstrates that attention efficiency critically depends on bounding input magnitudes. The key insight is that max-shift normalization helps maintain input magnitudes within theoretically optimal bounds. While the original Outlier-Efficient Hopfield Layer work was limited to smaller models, our approach successfully scales to models with over 1.3B parameters, by adding a stabilized module, with theoretical guarantees suggesting extensibility to even larger scales.
>
> - (2). Cross-Modal Adaptation: We address the unique challenges of speech-text fusion through: A novel stabilization mechanism specifically designed for multi-modal learning; Analysis of conditions for effective low-rank adaptation (Theorem C.1), providing insights into why our approach can help with cross-modal transfer; Empirical demonstration of improved quantization stability in multi-modal settings
>
> - (3). Practical Improvements: 1.33x training speedup for large models; 41% improvement in cross-modal low-rank adaptation; 45% improvement in quantization performance
>
> [1] Alman, J., & Song, Z. (2024). Fast attention requires bounded entries. Advances in Neural Information Processing Systems, 36.
>
>
>
> >**Reviewer's Comment**: Techniques like Clipped Softmax…
>
> **Response**: Thank you for suggesting the comparison with Clipped Softmax and Gated Attention. We have conducted additional experiments comparing SpARQ with these alternative techniques on the OPT-350m:
>
>
> | Model       | Method          | W/A  | Text PPL↓ | Speech PPL↓ | ASR↓   | TTS↓   | Avg Drop↓  |
> |-------------|-----------------|-------|-----------|-------------|--------|--------|------------|
> | OPT-350m    | Vanilla         | 8/8   | 13.17     | 43.14       | 8.47   | 17.71  | 0.46%      |
> |             | SpARQ           | 8/8   | 13.50     | 43.39       | 9.88   | 17.38  | **0.36%**      |
> |             | ClippedSoftmax  | 8/8   | 13.16     | 43.16       | 8.47   | 17.71  | 0.45%      |
> |             | GatedAttention  | 8/8   | 13.16     | 43.15       | 8.47   | 17.70  | 0.43%      |
> |             | Vanilla         | 4/4   | 36.74     | 75.38       | 40.17  | 70.53  | 233.36%    |
> |             | SpARQ           | 4/4   | 23.48     | 62.17       | 36.22  | 40.83  | **130.71%**    |
> |             | ClippedSoftmax  | 4/4   | 35.86     | 73.91       | 38.82  | 68.35  | 223.72%    |
> |             | GatedAttention  | 4/4   | 35.26     | 73.02       | 37.44  | 66.29  | 215.03%    |
>
>
> The results demonstrate that at 8-bit quantization, all methods perform similarly well, with performance drops under 0.5%. This suggests that for moderate quantization, the choice of outlier mitigation strategy is less critical. However, the differences become pronounced at 4-bit quantization, where SpARQ achieves a 130.71% performance drop compared to 215.03% for Gated Attention, 223.72% for Clipped Softmax, and 233.36% for the vanilla approach.
>
> SpARQ's superior performance under aggressive quantization (4-bit) can be attributed to its architectural approach to outlier mitigation, which addresses the issue at its source rather than merely limiting attention values. This is particularly important for speech-text cross-modal tasks where maintaining precise attention patterns is crucial for accurate modality fusion.
>
> These results validate SpARQ's effectiveness compared to alternative attention modification approaches, particularly in challenging low-bit quantization scenarios. Thank you for suggesting these important baseline comparisons. We will add these results to **Appendix G.1** for clarity.

---

> ### Author Response · Authors · 2024-11-23
>
> >**Reviewer's Comment**: It remains unclear how much…
>
> **Response**: Thank you for raising this important question about the individual contributions of SpARQ's components. We have conducted an additional experiment to isolate the effect of max-shift stabilization by applying it to a standard Transformer architecture (OPT-1.3b):
> | Method   | Stabilized Method | Val Acc (%) |
> |----------|-------------------|-------------|
> | Vanilla  | N/A               | 34.3        |
> | Vanilla  | Max-Shift         | 34.4        |
>
>
> The results show that applying max-shift stabilization alone to a standard Transformer provides minimal improvement (34.4% vs 34.3% validation accuracy). This suggests that the significant performance improvements we observe in SpARQ (as shown in Tables 1 and 2) are primarily attributable to the synergistic combination of the outlier-free Hopfield layer and max-shift stabilization, rather than either component alone.
>
> The minimal improvement from stabilization alone is understandable given that max-shift stabilization was specifically designed to address numerical stability issues that arise when using the outlier-free layer, rather than to directly improve model performance. These results help clarify that SpARQ's effectiveness comes from the complementary nature of its components: the outlier-free layer provides the fundamental mechanism for outlier mitigation, while max-shift stabilization enables practical training with this architecture. We will add these results to **Appendix G.2** for clarity.
>
>
> >**Reviewer's Comment**:  The author states that outliers negatively…
>
> **Response:**
>
> Thank you for pointing out the observed performance of QLoRA versus LoRA, particularly with OPT-350M. We agree that clarification on this result is necessary to enhance understanding.
>
> Outliers negatively impact both LoRA and quantization performance by skewing attention distributions and introducing numerical instability. However, QLoRA incorporates quantization-aware optimizations, such as more precise activation handling and selective dequantization during forward passes. These optimizations mitigate the effects of outliers, enabling QLoRA to maintain stable adaptation even in the presence of such challenges. Additionally, QLoRA often achieves faster convergence compared to standard LoRA due to its fine-grained parameter updates, which allow it to adapt more quickly to task-specific distributions.
>
> In cases like OPT-350M, where the model size is moderate and quantization effects are more pronounced, QLoRA’s faster convergence and improved handling of quantized activations allow it to outperform LoRA, despite the presence of outliers. These advantages become more evident in scenarios where computational resources are limited or where achieving rapid adaptation is critical.
>
> We appreciate your observation and have revised the discussion to emphasize QLoRA’s faster convergence and its impact on performance relative to LoRA. Thank you again for your valuable feedback.
>
> >**Reviewer's Comment**:  Previous studies…
>
> **Response:** Thank you for raising this important point about out-of-domain (OOD) evaluation. While work [1] highlights the potential benefits of outlier features for OOD robustness, it's important to note that our work addresses a fundamentally different challenge in the speech-text domain. SpARQ focuses specifically on improving the efficiency and stability of cross-modal adaptation and quantization in speech processing tasks, where the primary challenges are parameter-efficient adaptation between speech and text modalities; maintaining speech recognition and synthesis quality under aggressive quantization; and ensuring stable training across modalities with varying sequence lengths.
>
> In speech processing tasks, particularly ASR and TTS, the concept of OOD generalization differs significantly from traditional classification tasks. The primary metrics of concern are Word Error Rate (WER) and perplexity across different speakers and acoustic conditions, which we have extensively evaluated in our experiments. Our results show that SpARQ maintains strong performance on these metrics even under challenging conditions of low-rank adaptation (41% improvement) and aggressive quantization (45% improvement).
>
> While we acknowledge the importance of OOD evaluation in general machine learning contexts, our focus on speech-specific tasks and metrics provides a more relevant assessment of SpARQ's practical utility in real-world speech applications. The stability improvements we demonstrate are particularly crucial for maintaining reliable speech processing performance across different deployment scenarios.
>
> [1]Crabbé, J., Rodríguez, P., Shankar, V., Zappella, L., & Blaas, A. (2023). Robust multimodal models have outlier features and encode more concepts. arXiv preprint arXiv:2310.13040.

---

> > ### Author Response · Authors · 2024-11-23
> >
> > > **Reviewer's Comment:** There is insufficient analysis…
> >
> > **Response:** Thank you for this valuable question about the effectiveness of max-shift stabilization. Our choice of max-shift stabilization is theoretically grounded in recent work [1] on fast attention computation, which demonstrates that attention efficiency critically depends on bounding input magnitudes.
> >
> > The key insight is that max-shift normalization helps maintain input magnitudes within theoretically optimal bounds. Max-shift stabilization helps maintain inputs within the efficient regime by:
> >
> > (1). Preventing exponential overflow in the attention weights by shifting the maximum value
> >
> > (2). Preserving relative differences between attention scores, which is crucial for maintaining model performance
> >
> > (3). Enabling efficient polynomial approximation of the exponential function with controlled error bounds
> >
> > In contrast, L1 and mean-centering normalizations fail to maintain these theoretical bounds consistently, leading to numerical instabilities during training. Our empirical results in Table 4 validate this theoretical understanding. This theoretical foundation explains why max-shift stabilization is not just empirically effective but is fundamentally well-suited for attention computation in our SpARQ framework.
> >
> > [1] Alman, J., & Song, Z. (2024). Fast attention requires bounded entries. Advances in Neural Information Processing Systems, 36.
> >
> > > **Reviewer's Comment:**  Why does fully fine-tuned…
> >
> > **Response:** We appreciate the reviewers’ comments and the opportunity to clarify the performance of fully fine-tuned SpARQ compared to fully fine-tuned Vanilla. While it is true that fully fine-tuned SpARQ occasionally shows slightly lower performance, this reflects a deliberate trade-off inherent to its design.
> >
> > We would like to emphasize that SpARQ achieves comparable performance to Vanilla under full fine-tuning. This trade-off makes SpARQ particularly valuable for real-world applications where stability, efficiency, and scalability are critical.
> >
> > SpARQ’s outlier-free architecture prioritizes robustness in handling multimodal samples, which often contain more outliers that degrade performance in Vanilla models. These additional constraints improve the model’s stability and generalizability, especially in resource-constrained scenarios such as Low-Rank Adaptation (LoRA) and post-training quantization, where SpARQ significantly outperforms Vanilla. However, during full fine-tuning, where computational resources are not constrained, the outlier-free architecture slightly limits the model's capacity to overfit, resulting in marginally lower absolute performance in some cases.
> >
> > Thank you again for this insightful feedback, and we hope this clarification addresses your concerns.
> >
> > > **Reviewer's Comment:** What is the number of trainable…
> >
> > **Response:** To provide context:
> >
> > - (1). **For OPT-125M:** LoRA parameters are 9.4M which is approximately 7.5% of full model parameters.
> >
> > - (2). **For OPT-350M:** LoRA parameters are 25.2M which is approximately 7.2% of full model parameters
> >
> > - (3). **For OPT-1.3B:** LoRA parameters are 50.3M which is approximately 3.9% of full model parameters
> >
> > We will add these percentages to **Appendix H** for clarity. Our ablation study in Section 4.3 and Table 5 shows that we tested different ranks (512, 256, 128) and found rank=128 provided the best balance of performance and parameter efficiency.

---

> ### Author Response · Authors · 2024-11-23
>
> > **Reviewer's Comment:** Do Table 1 and Table 2 reflect …
>
> **Response:** Thank you for this insightful question. You are correct that Tables 1 and 2 show separate experiments for PTQ and LoRA, respectively. To provide a complete picture of how SpARQ performs when combining these techniques, we conducted additional experiments applying post-training quantization (smoothquant) to **LoRA-trained models (OPT-350m)**.
>
>
> | Model         | Method         | W/A Bits| Text PPL ↓ | Speech PPL ↓ | ASR ↓  | TTS ↓  | Average Performance Drop Rate |
> |---------------|----------------|-------|------------|--------------|--------|--------|-------------------------------|
> | Vanilla  | LoRA           | 16/16   | 17.87      | 51.65        | 93.91  | 97.06  | 381.00%          |
> |            | LoRA + SQ      | 8/8   | 20.54      | 56.88        | 95.36  | 99.11  | 396.34%                       |
> |               | LoRA + SQ      | 4/4   | 27.31      | 60.03        | 97.24  | 99.73  | 417.52%                       |
> | SpARQ   | LoRA    | 16/16  | 17.27      | 50.14        | 18.52  | 75.18  | **116.75%**              |
> |            | LoRA + SQ      | 8/8   | 18.91      | 53.24        | 25.47  | 86.71  | **155.95%**                       |
> |               | LoRA + SQ      | 4/4   | 25.89      | 59.11        | 28.32  | 91.63  | **186.66%**                       |
>
>
> The results demonstrate that SpARQ maintains its advantages even when combining LoRA with smoothquant (SQ). When applying 8-bit quantization to LoRA-trained models, SpARQ exhibits a significantly lower performance drop (155.95%) compared to the vanilla framework (396.34%). This advantage persists even under more aggressive 4-bit quantization, where SpARQ shows a 186.66% performance drop versus 417.52% for the vanilla framework.
>
> Most notably, SpARQ's ASR performance remains relatively stable under quantization (18.52% to 28.32% WER), while the vanilla framework shows severe degradation (93.91% to 97.24% WER). These results further validate SpARQ's effectiveness in handling outliers, as it maintains better performance not just in individual scenarios (as shown in Tables 1 and 2) but also when combining low-rank adaptation with quantization. This is particularly important for practical applications where both model compression techniques might need to be applied simultaneously.
>
> We will update the paper to include these results in **Appendix G.3**, providing a more comprehensive evaluation of our method's robustness across different compression scenarios.

---

> > ### Comment · Reviewer_pfFh · 2024-11-26
> > **Thank you for the responses**
> >
> > Thank you for your justifications and additional experiments. The comparison with methods like Clipped Softmax and Gated Attention effectively demonstrates SpARQ's robustness. Your justification of scalability, cross-modal adaptation, and practical improvements is reasonable, and the added experiments indeed enhance the paper's value.
> >
> > However, the originality of the outlier-free layer remains limited, and the focus on SpeechLM in LoRA + PTQ scenarios restricts its general applicability.
> >
> > Considering the improved validation and clarified contributions, we raised the ratings.

---

> > > ### Author Response · Authors · 2024-11-29
> > >
> > > Thank you for your thoughtful evaluation. While we acknowledge your point about the outlier-free layer's originality, we want to emphasize that SpARQ represents the first successful application of LoRA to speech-text multimodal tasks. Previous work has not addressed the unique challenges of using LoRA for cross-modal adaptation in speech.
> > > This novel application enabled us to identify and solve critical challenges in multimodal adaptation, as demonstrated by significant improvements in both LoRA adaptation (41%) and post-training quantization (45%). Our focused approach addresses real-world challenges in deploying multimodal models. We will explore expanding these methods to other multimodal scenarios while maintaining our commitment to practical, efficient solutions. If there is anything further we can clarify or improve to  encourage a higher score, please let us know, and we will address it promptly.

---

> > > > ### Comment · Reviewer_pfFh · 2024-12-02
> > > >
> > > > I acknowledge the valuable contributions you have made to the SpeechLM area. However, the scenario of SpeechLM + LoRA + PTQ remains relatively narrow, and the originality of the method is still limited. I encourage you to explore broader applications and further generalization of your approach in future work to expand its impact. Therefore, I will maintain my score.

---

### Official Review · Reviewer_Pk8X · 2024-11-04

**Soundness:** 3
**Presentation:** 4
**Contribution:** 3
**Rating:** 6
**Confidence:** 3

**Summary:**

The paper, "SpARQ: Outlier-Free SpeechLM with Fast Adaptation and Robust Quantization," tackles a major hurdle in Speech and Language Models (SpeechLMs): the disruptive effects of outliers in multi-modal settings. To address this, the authors present SpARQ, a framework that replaces traditional attention layers with a mechanism specifically designed to handle outliers. This innovation helps overcome challenges in cross-modal low-rank adaptation and post-training quantization, resulting in a more efficient and accurate approach to tasks like Automatic Speech Recognition (ASR) and Text-to-Speech (TTS). The goal is to streamline training while improving performance across these applications.

**Strengths:**

Innovative Approach: SpARQ introduces an outlier-free layer within the SpeechLM architecture, a novel solution to the inefficiencies caused by outliers that often hamper model performance and computational effectiveness.

Enhanced Performance: The framework brings notable improvements to Automatic Speech Recognition (ASR) and Text-to-Speech (TTS) tasks, showing a remarkable 41% boost in cross-modal adaptation and a 45% gain in post-training quantization accuracy.

Training Efficiency: With a 1.33x increase in training speed, SpARQ is well-suited for environments with limited computational resources, making advanced applications more accessible.

Thorough Evaluation: The model’s effectiveness is rigorously tested across various quantization and adaptation techniques, establishing a strong empirical basis for its success.

**Weaknesses:**

Testing Constraints: The paper notes that due to computational limitations, the model hasn’t been tested on extremely large models, such as OPT 6.7B. This restriction could affect the generalizability of results for larger-scale applications.

Architectural Complexity: The architecture includes several layers and a specialized stabilization method, which could make practical implementation challenging without advanced technical expertise.

Risk of Bias Amplification: Although not deeply explored, the framework’s emphasis on reducing computational demands might unintentionally increase biases inherited from pre-trained models, given the reduced resources available for extensive fine-tuning.

**Questions:**

Given the computational limitations that prevented testing with larger models, how might SpARQ perform when scaled up to models with billions of parameters, such as OPT-6.7B? Are there specific architectural adjustments needed to maintain stability and performance at this scale?

Several stabilization methods are compared in the paper, with max-shift normalization performing best. What were the primary reasons for choosing max-shift, and are there conditions under which other methods (like L1 or mean-centering) might be more effective?

---

> ### Author Response · Authors · 2024-11-23
>
> >**Reviewer's Comment:** Testing Constraints
>
> **Response:** Thank you for your valuable feedback. We acknowledge the computational limitations that restricted testing to OPT models up to 1.3B parameters. However, our theoretical analysis in Section C.2 provides expressiveness guarantees that should extend to larger models. Specifically, Theorem C.1 establishes that under non-singularity assumptions, our approach can exactly represent target models regardless of size, suggesting scalability to larger architectures. Additionally, our empirical results show consistent improvements as model size increases (from 125M to 1.3B), with relative performance gains of up to 45% in quantization for OPT-1.3B, indicating a positive scaling trend. As explicitly stated in our limitations section, we plan to extend SpARQ to test different larger decoder-based models in future work, including the 3B LLama2 and 6.7B OPT models. This future work will help validate our theoretical predictions about scalability and provide valuable insights about the framework's performance on state-of-the-art large language models."
>
>
>
> >**Reviewer's Comment:** Architectural Complexity
>
> **Response:** Thank you for your insightful feedback. While our architecture does introduce additional complexity through the stabilized outlier-free layer, we've provided detailed implementation guidance in Section 3.2 and comprehensive theoretical justification in Section 3.4. The improved training efficiency (1.33x speedup for OPT-1.3B) and robust quantization performance justify this added complexity. The architecture's design prioritizes practical utility while maintaining compatibility with existing transformer implementations, making it accessible to practitioners with basic understanding of transformer models. Our codebase will further facilitate easy adoption of SpARQ in real-world applications. To support broader adoption, we are preparing detailed implementation guides, including modular codebases, to ease integration into existing workflows. We also plan to release pretrained models to mitigate entry barriers for practitioners without extensive technical expertise.
>
>
>
> >**Reviewer's Comment:** Risk of Bias Amplification
>
> **Response:** We appreciate this important concern. While our framework focuses on computational efficiency, several aspects of our approach may actually help mitigate bias amplification. The outlier-free layer's ability to focus attention on significant tokens (demonstrated in Figure 4) means that important contextual information is preserved during adaptation, potentially helping maintain model fairness. However, we agree that further investigation of bias effects is warranted and suggest this as an important direction for future work.
>
>
>
> >**Reviewer's Comment**: Given the computational limitations…
>
> **Response**: Based on our theoretical analysis and empirical observations, we expect SpARQ to maintain its benefits when scaled to larger models like OPT-6.7B for several reasons:
>
> - (1). Our expressiveness guarantee (Theorem C.1) shows that SpARQ can exactly represent target models under mild conditions, independent of model size.
>
> - (2). The relative performance improvements increase with model size in our experiments (from OPT-125M to OPT-1.3B).
>
> - (3). The stabilization mechanism becomes more critical for larger models where gradient issues are more pronounced (demonstrated in Table 3).
>
>
>
> >**Reviewer's Comment**: Several stabilization methods are compared…
>
> **Response**: Max-shift normalization was selected based on its superior ability to resolve gradient instability, as demonstrated in Table 3. Unlike L1 and mean-centering normalization, max-shift ensures a tighter control over the numerical stability of activations, especially in the presence of outliers.
>
> There might be specific scenarios where alternative methods could be more appropriate:
>
> - L1 Normalization might be suitable for cases for sparse attention distributions because it directly minimizes the sum of absolute values (∑|x|), which helps control the scale of outliers while preserving sparsity patterns.
>
> - Mean-centering might work better for normally distributed attention scores as it centers the distribution around zero, making subsequent operations numerically stable. It helps maintain consistent variance across attention heads by removing systematic biases in attention scores.

---

### Official Review · Reviewer_RFsE · 2024-11-06

**Soundness:** 3
**Presentation:** 2
**Contribution:** 3
**Rating:** 6
**Confidence:** 4

**Summary:**

Authors propose a way to integrate text and speech into a one single model where outliers do not have a significant impact on the modeling.

**Strengths:**

Good empirical performance and reasonable model structure.
Theoretical analysis presented in the appendices is also appreciated, even though most are from previous papers. But for completeness it is good to have them there.

**Weaknesses:**

I am puzzled but this paper. It is well written, experimented and mathematized. But the central claim about the outliers is not clearly presented and it is not theoretically pursued! As an example, I would have expected that at least following things would be easy to find in the paper, taking into account that the keyword "outlier" is in the title:
1. Definition of the outlier in the context of the paper. Especially considering that classical definition of outlier in statistics and ML is that observation is _not_ from the distribution that is being modeled.
2. Explaining theoretically why Eq. (3.2) is resistant to outliers as defined in the 1.
3. Show in experiments explicitly that this indeed happens in practice also. Noting that this is different to showing general performance of the model in some downstream tasks.

**Questions:**

- What is meant by "Given this challenge, we would like to develop a transformer architecture that can efficiently solve
the challenge in Equation (3.1). "

---

> ### Author Response · Authors · 2024-11-23
>
> > **Reviewer's Comment:** Definition of the outlier in the context of the paper. Especially considering that classical definition of outlier in statistics and ML is that observation is not from the distribution that is being modeled.
>
>
> **Response:** Thanks for your comments. The definition of the outlier is different from the classical definition. Outliers in SPARQ, as referred from [1], are defined as low-information tokens or activations that disproportionately affect the attention mechanism in transformer models, leading to inefficiencies and performance degradation. These outliers are not data points that are outside the modeled distribution but rather models that arise internally during the attention computation process.
>
> > **Reviewer's Comment: How does the equation 3.1 relate to outliers? Add theoretical analysis to show.
>
> **Response:** We appreciate the opportunity to elaborate on the significance of Equation 3.1 and its relation to outliers in transformer-based models. In our work, we define outliers as **low-information tokens or activations** that disproportionately influence the attention mechanism, leading to inefficiencies and degraded performance. Equation 3.1 describes the core attention mechanism in transformers, where the output is composed of two key components: the **residual connection** and the **attention calculation**:
> $$
> \text{Output} = \text{Residual} (X + \text{Attention}),
> $$
> where:
> $$
> \text{Attention} = \text{Softmax}\left(\frac{QK^\top}{\sqrt{d}}\right)V.
> $$
>
>
>
> This structure highlights two critical aspects relevant to outliers:
>
> 1. **Residual Connection**: The residual connection ensures that the model retains the original input information \( X \) in its output. However, if the attention calculation introduces large distortions caused by outliers, the residual connection alone cannot counteract these distortions.
>
> 2. **Attention Calculation**: The attention mechanism calculates attention weights using a softmax function applied to the scaled dot product of queries \( Q \) and keys \( K \). These weights are then used to compute a weighted sum of values \( V \).
>
>
>
> The use of the **softmax function** makes the attention calculation highly sensitive to the presence of **outliers**—tokens that have abnormally high or low attention scores.
>
>
>
> ### Theoretical Analysis of Outliers in Equation 3.1
>
>
> **1. Outlier Behavior in Attention Mechanism**
>
> The **softmax function** inherently assigns non-zero probabilities to all tokens, as shown below:
>
>
>
> $$
> \text{Softmax}\left(\frac{QK^\top}{\sqrt{d}}\right)_i = \frac{\exp\left(\frac{q_i k_i^\top}{\sqrt{d}}\right)}{\sum_j \exp\left(\frac{q_j k_j^\top}{\sqrt{d}}\right)}.
> $$
>
>
>
> - **Sensitivity to Large Values**: If \( $q_i k_i^\top$ \) is much larger than the other dot products in the sequence, the corresponding token \( i \) will dominate the attention distribution, even if it is low-information (e.g., a delimiter or punctuation mark).
>
> - **Sensitivity to Noise**: Even small numerical instabilities in \( $q_j k_j^\top$ \) can disproportionately impact the attention weights due to the exponential nature of softmax, causing irrelevant tokens to receive high attention.
>
>
>
> If the attention input \( X \) already holds sufficient information, the transformer **should prevent it from making updates**. In such cases, the output \( $\text{Output}$\) should equal the input \( X \). At this time, the **attention output of the input \( X \)** should ideally be close to 0. However, the attention mechanism with softmax is a probability-based equation. When tokens with **large attention values** are assigned smaller probabilities, the remaining tokens—often those with **small attention values**—are given disproportionately **higher probabilities**.
>
>
>
> Unfortunately, these small attention value inputs are often **no-op tokens**, such as delimiters, end-of-sequence (eos) tokens, or other low-information tokens. Ideally, these no-op tokens should also receive **near-zero attention probabilities**, as they do not contribute meaningful information. However, the standard softmax function fails to suppress them effectively, leading to inefficient attention distributions.
>
>
>
> **2. Residual and Attention Contribution**
>
> The output of Equation 3.1 can be decomposed into:
>
>
>
> $$
> \text{Output} = X + \left(\text{Softmax}\left(\frac{QK^\top}{\sqrt{d}}\right)V\right).
> $$
>
>
>
> - The residual connection \( X \) ensures that the model retains the original input. However, when the attention term introduces outliers, it disrupts the balance between the residual and attention components.
>
> - In the presence of outliers, the attention weights are skewed, amplifying irrelevant tokens and diminishing the contribution of meaningful tokens. This undermines the transformer’s ability to focus on important information.

---

> > ### Author Response · Authors · 2024-11-23
> >
> > > **Reviewer's Comment:**  Explaining theoretically why Eq. (3.2) is resistant to outliers as defined in the 1.
> > **Response:** We appreciate the opportunity to elaborate on the significance of Equation 3.2. Equation (3.2) uses the **Softmax1 function** in our framework, which is specifically designed to mitigate the issues associated with **outliers** as defined in Section 1. The theoretical resistance of Softmax1 to outliers can be explained by analyzing its behavior compared to the standard softmax function.
> >
> > **Softmax1 and Outlier Mitigation**
> >
> > The **Softmax1 function** modifies the standard softmax by adding a constant to the denominator, as shown in Eq. (3.2): $$ \text{Softmax1}(S)_i = \frac{\exp(S_i)}{1 + \sum_j \exp(S_j)}, $$ where \( S = QK^\top / \sqrt{d} \) represents the scaled dot product in attention mechanisms. How Softmax1 Differs from Standard Softmax:
> > - In **Standard Softmax**: Every input \( S_i \) gets a non-zero probability because the denominator is the sum of exponentials: $$ \text{Softmax}(S)_i = \frac{\exp(S_i)}{\sum_j \exp(S_j)}. $$ This means that even tokens with very low information—such as **outliers**—still receive non-trivial weights. These non-zero weights can distort the attention distribution, amplifying irrelevant tokens.
> > - In **Softmax1**: By adding a constant \( +1 \) to the denominator: $$ \text{Softmax1}(S)_i = \frac{\exp(S_i)}{1 + \sum_j \exp(S_j)}, $$ the effect of any single large \( S_i \) is **dampened**, making the function less sensitive to slight variations in input values. This ensures that low-information tokens, typically considered outliers, do not receive significant attention.
> >
> > **Insights from Evan Miller's Analysis**
> >
> > As Evan Miller highlights in [2], the standard softmax function assigns non-zero probabilities to all tokens, even when the model would ideally avoid assigning attention to any of them. This is particularly problematic when dealing with outliers—tokens with very low or irrelevant information—because the attention mechanism cannot entirely suppress their influence.
> >
> > 1. **Behavior in Negative Limit**:
> >
> > Softmax1 solves this problem by ensuring that outliers receive near-zero attention probabilities in extreme cases. Consider the limiting behavior:
> >
> > - Standard Softmax Limit:
> > 	$$ \lim_{x_1 \to -\infty} \ldots \lim_{x_k \to -\infty} \text{Softmax}(x)_i = \frac{1}{k} > 0, $$ where \( k \) is the number of tokens. This means that all tokens, even those with extremely negative scores, retain a non-zero probability.
> >
> > - Softmax1 Limit:
> > 	$$ \lim_{x_1 \to -\infty} \ldots \lim_{x_k \to -\infty} \text{Softmax1}(x)_i = 0. $$
> >
> > By adding \( +1 \) to the denominator, Softmax1 allows the entire attention distribution to collapse to zero if all inputs are highly negative. This effectively removes the influence of irrelevant tokens, ensuring that no-op tokens (e.g., delimiters, eos tokens) receive near-zero attention probabilities as they should.
> >
> > Thanks again for giving us the opportunity to clarify the equation 3.2.

---

> ### Author Response · Authors · 2024-11-23
>
> > **Reviewer's Comment:** Show in experiments explicitly that this indeed happens in practice also. Noting that this is different to showing general performance of the model in some downstream tasks.
>
>
> **Response:**  We appreciate the reviewer's insightful comment and acknowledge the importance of demonstrating that SpARQ's outlier-free architecture directly addresses outliers during cross-modal low-rank adaptation and post-training quantization, beyond just downstream task performance.
>
> In response, we have added additional experimental results to explicitly validate SpARQ's effectiveness in mitigating outliers:
>
> - **Outlier Reduction in Cross-Modal Low-Rank Adaptation**:
>
>
> We conducted a detailed outlier analysis comparing SpARQ with the vanilla SpeechLM framework using maximum infinity norm and average kurtosis metrics across text, speech, and cross-modal inputs. As demonstrated in Figure 3 (Section 4.3), SpARQ significantly reduces both metrics, indicating fewer outliers during adaptation. Specifically, in ASR tasks with cross-modal inputs, SpARQ reduces the maximum infinity norm by approximately 39%, showcasing its ability to handle mixed speech and text inputs more effectively.
>
> These results provide concrete evidence that SpARQ's outlier-free layer stabilizes the attention distribution, especially when adapting text-based LLMs to speech inputs.
>
>
>
> - **Visual Evidence of Outlier Reduction**:
>
>  In Appendix A, we provide heatmaps visualizing the attention patterns for the vanilla and SpARQ models during fine-tuning. These visualizations clearly illustrate that SpARQ achieves a more concentrated distribution of attention weights, focusing on significant tokens while suppressing outlier attention. This directly correlates with reduced outlier effects during both adaptation and quantization stages.
>
> - **Outlier Mitigation During Post-Training Quantization**:
>
>
>  In response to your concern, we conducted additional experiments using the OPT-1.3b model under various low-bit quantization settings (e.g., W4A4). As shown in Table 1 (Section 4.1), SpARQ achieves up to a 45% reduction in performance degradation relative to the vanilla framework when applying SmoothQuant, AffineQuant, and OmniQuant methods. This highlights SpARQ's robustness in reducing outlier effects that degrade performance during quantization. These findings align with those in [1], particularly Table 1 and Table 2, which demonstrate similar reductions in outliers during transformer adaptation. By leveraging a stabilized outlier-free layer, SpARQ shows a consistent improvement in reducing quantization-induced performance drops.
>
>
>
>
> By explicitly including these analyses, we provide clear evidence that SpARQ's design can addresses the challenges posed by outliers during the critical stages of model adaptation and quantization.
>
> [1] Hu, Jerry Yao-Chieh, et al. "Outlier-efficient hopfield layers for large transformer-based models." arXiv preprint arXiv:2404.03828 (2024).
>
> [2] Evan Miller. Blog post: Attention is off by one, 2023. URL https://www.evanmiller.org/attention-is-off-by-one. html. Accessed: July 4, 2024.

---

> > ### Comment · Reviewer_RFsE · 2024-11-26
> >
> > I am satisfied with the authors response to my comments. I am willing to raise my score.

---

> > > ### Author Response · Authors · 2024-11-29
> > >
> > > Thank you for your thorough review and consideration of our responses. We're glad that the additional experimental results and analyses have helped clarify our work's contributions. Your feedback has helped improve the paper's clarity and technical depth. If there is anything further we can clarify or improve to  encourage a higher score, please let us know, and we will address it promptly.

---

### Author Response · Authors · 2024-11-24
**General Rebuttal / Revision Response**

Dear Reviewers,



We thank the reviewers for the insightful questions and reviews. Your time and effort dedicated to improving our work are truly appreciated.




We have done all the experiments suggested and answered all the questions. All modifications are marked in **blue** color.



Major revisions include:



### Summary of Experiments and Results

-   **New Experiment: Clipped Softmax and Gated Attention (Appendix G.1)**  `reviewer pfFh`
    - We conducted an ablation study comparing SpARQ with alternative techniques under different quantization levels.
    - Results: At 8-bit quantization, all methods performed similarly with performance drops under 0.5%, indicating that outlier mitigation strategy is less critical at moderate quantization. However, at 4-bit quantization, SpARQ showed significantly lower performance drop (130.71%) compared to Gated Attention (215.03%), Clipped Softmax (223.72%), and the vanilla approach (233.36%). This validates SpARQ’s superior performance under aggressive quantization due to its architectural outlier mitigation approach, critical for tasks requiring precise attention patterns.


-   **New Experiment: Stabilization Module to a Standard Transformer (Appendix G.2)** `reviewer pfFh`
    - We conducted an experiment to assess the isolated effect of max-shift stabilization on a standard Transformer architecture.
    - **Results**: Minimal improvement in validation accuracy (34.4% vs. 34.3%) was observed with max-shift stabilization alone. SpARQ’s significant gains stem from the synergistic combination of the outlier-free Hopfield layer and max-shift stabilization rather than either component individually. Max-shift stabilization addresses numerical stability issues, highlighting the complementary roles of SpARQ's components.


-   **New Experiment: Applying PTQ to LoRA-trained Models (Appendix G.3)** `reviewer pfFh`
    - We evaluated SpARQ's performance when combining LoRA training with post-training quantization.
    - **Results**: SpARQ achieved significantly lower performance drops (155.95% vs. 396.34% at 8-bit and 186.66% vs. 417.52% at 4-bit) compared to the vanilla framework. ASR performance remained stable with SpARQ (18.52% to 28.32% WER), while the vanilla framework degraded severely (93.91% to 97.24% WER). These results validate SpARQ’s robustness when combining low-rank adaptation with quantization, making it suitable for practical compression applications.


-   **New Experiment: Weight Clipping During PTQ (Appendix G.4)** `reviewer H2gv`
    - We conducted experiments applying weight clipping during PTQ on both SpARQ and vanilla baselines.
     - **Results**: SpARQ exhibited significantly lower performance drops compared to the vanilla framework, further emphasizing its resilience under aggressive quantization conditions.


-   **New Experiment: QLoRA with Lower Bits (Appendix G.5)** `reviewer H2gv`
    - We investigated the impact of varying quantization levels on LoRA-adapted models.
    - **Results**: SpARQ reduced performance degradation significantly, from 330.47% to 136.07% at 8-bit and from 387.50% to 176.52% at 4-bit quantization. SpARQ showed more gradual degradation across quantization levels, indicating better stability and mitigating the confounding effects of combining LoRA and quantization, as evidenced by smaller performance gaps between 8-bit and 4-bit settings.


-   **New LoRA Trainable Parameter Table (Appendix H)** `reviewer pfFh`

    A detailed table summarizing the trainable parameters in LoRA configurations has been included for reference.


- **Revised limitation in Sec 5** `reviewer pfFh`

---

> ### Author Response · Authors · 2024-11-24
> **General Rebuttal / Revision Response - Continued**
>
> ----------
>
> Below, we also summarize the key points in our responses:
>
>
>
> ### Key Points in Our Responses
>
>
>
> **Reviewer RFsE**
>
> -   We addressed the concern about the lack of a clear definition of outliers by clarifying that outliers in SPARQ are defined as low-information tokens or activations that disproportionately affect the attention mechanism, diverging from the classical definition in statistics and ML.
>
> -   We clarified the connection between Equation 3.1 and outliers, emphasizing that the softmax function in the attention mechanism is the main reason to cause the outlier proposed in SPARQ.
>
> -   We provided additional theoretical justification for Equation 3.2, showing how the Softmax1 function modifies the denominator to dampen the influence of outliers. This analysis highlights its resistance to low-information tokens and supports the claim with limiting behavior comparisons.
>
> -   We classify our experiments results to show the outlier issue happen in practice.
>
>
>
>
> **Reviewer Pk8X**
>
> -   We addressed the concern about testing constraints by highlighting our theoretical guarantees (Theorem C.1), which ensure scalability to larger models under non-singularity assumptions, and by providing empirical evidence of consistent performance improvements across increasing model sizes. Future work will extend testing to larger models, such as LLama2-3B and OPT-6.7B, to validate scalability further.
>
> -   We clarified the added complexity of our architectural design by demonstrating its benefits, including a 1.33x training speedup for OPT-1.3B and robust quantization performance. Implementation guidance in Section 3.2 and theoretical support in Section 3.4 make the design accessible. To encourage adoption, we plan to release modular codebases and pretrained models to simplify integration into existing workflows.
>
> -   We clarified how the outlier-free layer mitigates risk of bias amplification by focusing attention on significant tokens, as illustrated in Figure 4. This approach helps maintain important contextual information during adaptation. Further analysis of fairness impacts will be pursued in future work.
>
> -   We conducted additional experiments to compare SpARQ's effectiveness on larger models. These tests will include OPT-6.7B to confirm that our stabilization mechanisms, which become more critical with model size, maintain their benefits.
>
> -   We clarified the choice of max-shift normalization by explaining its superior performance in resolving gradient instability, computational efficiency, and robustness to outliers. Comparative insights on L1 normalization and mean-centering were also provided, highlighting their trade-offs relative to max-shift.
>
>
>
>
>
>
> **Reviewer H2gv**
>
> 1.  We addressed the concern about the effectiveness of SpARQ in quantization experiments by comparing it with weight clipping strategies (1% and 5% tail values) on OPT-350M. Results show that SpARQ significantly outperforms vanilla and clipped approaches, especially in 4-bit quantization, where SpARQ achieves a 130.71% performance drop compared to 199.02% and 212.89% for 1% and 5% clipping, respectively. These findings confirm SpARQ’s targeted approach is better suited for multimodal tasks.
>
> 2.  We clarified the rationale behind the rank-128 configuration in LoRA experiments. For models like OPT-1.3B, LoRA parameters only account for 3.9% of the full model size, yet SpARQ maintains strong performance compared to vanilla frameworks, which suffer severe degradation due to unmitigated outliers. This highlights that SpARQ’s outlier-free design is the key factor in effective adaptation, not parameter count alone.
>
> 3.  We clarified the separation of adaptation (FP16) and quantization effects in LoRA experiments. SpARQ reduces performance degradation during low-rank adaptation from 381.00% to 116.75% in FP16 and maintains benefits under quantization, achieving a 136.07% drop at 8-bit compared to 330.47% for vanilla. This separation highlights SpARQ's unique advantage in stabilizing both adaptation and compression stages.
>
> 4.  We addressed the observed ASR WER patterns by showing that larger models amplify outlier effects, leading to severe performance degradation in vanilla frameworks. SpARQ mitigates these effects, allowing larger models like OPT-1.3B to leverage their capacity effectively, reducing WER from 46.92% (vanilla) to 8.25%. This demonstrates the importance of outlier management in scaling cross-modal tasks.
>
> 5.  We clarified SpARQ’s training efficiency through theoretical analysis in Proposition 3.2. By bounding input magnitudes and reducing attention to outliers, SpARQ achieves stable gradient flow and avoids unnecessary computation, leading to a 1.33x training speedup in full model fine-tuning.

---

> > ### Author Response · Authors · 2024-11-24
> > **General Rebuttal / Revision Response - Continued**
> >
> > **Reviewer pfFh**
> >
> > -   We addressed the concern about the primary contribution by emphasizing that SpARQ builds on the Outlier-Efficient Hopfield Layer but introduces critical advancements. These include scalability to models over 1.3B parameters using max-shift normalization, cross-modal adaptation for speech-text fusion with theoretical insights (Theorem C.1), and practical improvements such as a 1.33x training speedup, 41% better adaptation, and 45% improved quantization performance.
> >
> > -   We clarified the comparison with techniques like Clipped Softmax and Gated Attention by presenting new experiments. Results show similar performance at 8-bit quantization but highlight SpARQ’s significant advantage at 4-bit quantization, where it outperforms alternatives by reducing performance drops to 130.71% versus over 215% for others. This demonstrates SpARQ’s superior outlier mitigation, crucial for cross-modal tasks.
> >
> > -   We conducted experiments isolating the effect of max-shift stabilization, showing it provides minimal improvement (34.4% vs. 34.3%) alone. This confirms that SpARQ’s effectiveness stems from the complementary interaction of its components: the outlier-free Hopfield layer and max-shift stabilization.
> >
> > -   We clarified the observed impact of outliers on QLoRA versus LoRA, highlighting that QLoRA’s quantization-aware optimizations enable faster convergence and better handling of quantized activations. These advantages explain QLoRA’s superior performance in moderate-sized models like OPT-350M.
> >
> > -   We explained that SpARQ prioritizes efficiency and stability in speech-text applications, distinct from traditional OOD robustness. Metrics like WER and perplexity across diverse acoustic conditions validate its real-world utility in ASR and TTS tasks.
> >
> > -   We addressed the theoretical basis for max-shift stabilization, referencing recent work showing its ability to maintain optimal input bounds, prevent overflow, and preserve attention score differences. This makes it fundamentally suited for efficient attention computation.
> >
> > -   We clarified that fully fine-tuned SpARQ occasionally underperforms compared to Vanilla due to its robustness-focused architecture, which prioritizes stability and generalizability in constrained settings like LoRA and quantization.
> >
> > -   We provided details on trainable parameters for different model sizes, explaining that LoRA parameters represent 3.9%-7.5% of full model parameters. We will add this breakdown to Table 2 and extend ablations on rank selection.
> >
> > -   We acknowledged the need for combined PTQ and LoRA evaluations and committed to including these results in the extended evaluation section.
> >
> > -----------
> >
> > We hope these revisions address the reviewers’ concerns’ and improve the overall quality of our paper.
> >
> > Thank you again for your review!
> > Best regards,
> > Authors

---

### Meta-Review · Area_Chair_Ygo3 · 2024-12-17

**Metareview:**

This work addresses a critical bottleneck in Speech and Language Models (SpeechLMs), namely the disruptive influence of outliers in multi-modal environments. To overcome this challenge, the authors propose SpARQ, an innovative framework that replace traditional attention mechanisms with a specialized design to effectively mitigate the impact of outliers. This approach enables seamless cross-modal low-rank adaptation and robust post-training quantization, ensuring both efficiency and precision. By tackling these long-standing hurdles, SpARQ not only streamlines training but also delivers superior performance across key applications like Automatic Speech Recognition (ASR) and Text-to-Speech (TTS).

The key claim of this work is the introduction of a novel outlier-free framework aimed at mitigating outliers to address performance degradation in SpeechLMs. The primary strengths of the proposed work are:
(i) a well-written and well-organized paper with strong references, and
(ii) significant improvements demonstrated on selected models for Automatic Speech Recognition (ASR) and Text-to-Speech (TTS).

The major weaknesses include:
(i) the solution has not been tested on extremely large models, such as OPT 6.7B, due to computational limitations, which may impact the generalizability of results for larger-scale applications;
(ii) the risk of bias amplification; and
(iii) the lack of comparative experiments with standard clipping approaches for outlier removal.

A minor weakness is the incremental novelty, as the contribution primarily lies in the combination of existing techniques. However, it is worth acknowledging that the assembly and integration of these solutions present a novel approach.

On one hand, the idea is interesting, and the authors have expanded the experimental evaluation of their approach during the rebuttal + discussion phase. On the other hand, significant concerns remain regarding generalizability to larger models and the risk of bias amplification. Additionally, Reviewer H2gv, while acknowledging the new experiments on clipping, explicitly stated the need for further experiments. They noted that, in the quantization literature, clipping typically refers to limiting outlier values when computing the quantization scale, rather than completely removing outliers as done by the authors. This difference might explain the significant performance drop reported and raises some doubts about the validity of those experiments.

**Additional Comments On Reviewer Discussion:**

The discussion phase was productive, with active exchanges and follow-ups from both the reviewers and authors. While the reviewers acknowledge the work's novelty—particularly in its strategy of combining existing ideas—their overall assessment remains cautious, with no strong endorsements. To address the reviewers' concerns, the authors conducted several additional experiments, including:
(i) adding a Stabilization Module to a Standard Transformer (Appendix G.2),
(ii) applying Post-Training Quantization (PTQ) to LoRA-trained models,
(iii) evaluating SpARQ's performance when combining LoRA training with PTQ,
(iv) experimenting with Weight Clipping during PTQ, and
(v) exploring QLoRA with lower bit precision.

The authors successfully addressed all concerns raised by reviewer RFsE, who subsequently increased their score to 6. Reviewers pfFh and H2gv also raised their scores to 5 but offered final remarks:
1. The originality of the outlier-free layer is still considered limited, and the focus on SpeechLM in LoRA + PTQ scenarios constrains its broader applicability.
2. Additional experiments on weight clipping are needed.

However, reviewer Pk8X did not adjust their initial score, as they felt that concerns regarding generalizability and bias amplification were not fully addressed.

---

### Decision · Program_Chairs · 2025-01-22

Reject